# Epidemiology and Ecology of Influenza A Viruses among Wildlife in the Arctic

**DOI:** 10.3390/v14071531

**Published:** 2022-07-13

**Authors:** Jonathon D. Gass, Hunter K. Kellogg, Nichola J. Hill, Wendy B. Puryear, Felicia B. Nutter, Jonathan A. Runstadler

**Affiliations:** 1Department of Infectious Disease and Global Health, Cummings School of Veterinary Medicine, Tufts University, North Grafton, MA 01536, USA; hkkellogg@gmail.com (H.K.K.); wendy.puryear@tufts.edu (W.B.P.); felicia.nutter@tufts.edu (F.B.N.); jonathan.runstadler@tufts.edu (J.A.R.); 2Department of Biology, University of Massachusetts, Boston, MA 02125, USA; nichola.hill@umb.edu

**Keywords:** influenza A virus, Arctic, Subarctic, wild bird, transmission, climate, interhemispheric transmission, inter-continental transmission, reassortment

## Abstract

Arctic regions are ecologically significant for the environmental persistence and geographic dissemination of influenza A viruses (IAVs) by avian hosts and other wildlife species. Data describing the epidemiology and ecology of IAVs among wildlife in the arctic are less frequently published compared to southern temperate regions, where prevalence and subtype diversity are more routinely documented. Following PRISMA guidelines, this systematic review addresses this gap by describing the prevalence, spatiotemporal distribution, and ecological characteristics of IAVs detected among wildlife and the environment in this understudied region of the globe. The literature search was performed in PubMed and Google Scholar using a set of pre-defined search terms to identify publications reporting on IAVs in Arctic regions between 1978 and February 2022. A total of 2125 articles were initially screened, 267 were assessed for eligibility, and 71 articles met inclusion criteria. IAVs have been detected in multiple wildlife species in all Arctic regions, including seabirds, shorebirds, waterfowl, seals, sea lions, whales, and terrestrial mammals, and in the environment. Isolates from wild birds comprise the majority of documented viruses derived from wildlife; however, among all animals and environmental matrices, 26 unique low and highly pathogenic subtypes have been characterized in the scientific literature from Arctic regions. Pooled prevalence across studies indicates 4.23% for wild birds, 3.42% among tested environmental matrices, and seroprevalences of 9.29% and 1.69% among marine and terrestrial mammals, respectively. Surveillance data are geographically biased, with most data from the Alaskan Arctic and many fewer reports from the Russian, Canadian, North Atlantic, and Western European Arctic. We highlight multiple important aspects of wildlife host, pathogen, and environmental ecology of IAVs in Arctic regions, including the role of avian migration and breeding cycles for the global spread of IAVs, evidence of inter-species and inter-continental reassortment at high latitudes, and how climate change-driven ecosystem shifts, including changes in the seasonal availability and distribution of dietary resources, have the potential to alter host–pathogen–environment dynamics in Arctic regions. We conclude by identifying gaps in knowledge and propose priorities for future research.

## 1. Introduction

Global dissemination of influenza A viruses (IAVs) is driven by the migratory ecology of wild aquatic birds [1,2,3]. Reservoir hosts of IAVs include *Anseriformes* (ducks, geese, and swans) and *Charadriiformes* (gulls, terns, auks, and other shorebirds), many of which migrate biannually between southern wintering grounds and northern breeding grounds within Subarctic, low Arctic, and high Arctic regions (herein “Arctic regions”) (Figure 1) [3,4,5,6,7]. Previous research has implicated circumpolar Arctic regions as ecologically significant for the persistence and geographic spread of IAVs by wild avian hosts [8,9]. Wild bird breeding ranges in the Arctic facilitate inter-species transmission and genetic exchange between and among high and low pathogenic IAVs (HPAI and LPAI), particularly given the density of immuno-naïve juveniles in northern regions [9,10,11,12,13]. Both low temperature and annual rainfall have been associated with long-term IAV environmental stability and persistence in northern climates and temperate zones [14,15]. Arctic marine and terrestrial mammals are also susceptible to IAV infection, largely through seasonal reinfection from avian and environmental sources [16]. Despite evidence demonstrating the significance of northern regions for the migratory connectivity and global dissemination of IAVs [5], there is a dearth of consolidated data on the epidemiology and ecology of IAVs among wildlife in the circumpolar Arctic region. 

IAVs are enveloped, single-stranded, and negative sense RNA viruses which comprise eight genome segments, two of which code for the glycoproteins hemagglutinin (HA) and neuraminidase (NA), whose antigenic properties are used to classify viral sub-types [17]. These surface proteins are antigenically diverse due to their evolutionary interaction with selective pressures of the host immune system [18,19,20]. IAVs are classified as HPAI or LPAI based on the presence of a poly-basic proteolytic excision site between HA1 and HA2 subunits of the HA protein and its association with mortality in poultry [21,22]. To date, 18 HAs and 11 NAs have been discovered and almost all known subtype combinations have been isolated from wild birds, many of which migrate through Arctic regions [23]. While not all H5 and H7 subtypes contain poly-basic sequences, only H5 and H7 subtypes have proven to be highly pathogenic in wild birds [24,25]. All human influenza pandemics that can be traced to their origins (1918, 1957, 1968, 2009) emerged from wild birds, often facilitated by intermediate hosts such as swine and poultry [26,27]. Amplification of IAVs has been well characterized at the domestic animal–human interface; however, much less is understood about how interfaces between wild birds, other wildlife species, and environmental sources of transmission maintain animal infections in the natural world [28,29,30,31,32].

Given the phenology of Arctic regions for many migratory avian species and alterations to climate regimes due to global climate change (GCC), where warming trends are more than twice the global mean [33], it is vitally important to critically examine the literature on the ecology of IAVs among wildlife in Arctic regions. Primary epidemiologic and ecological data collection related to IAVs in this region is relatively sparse compared to other global regions, and the bulk of the literature on this topic is predominantly perspective, expert opinion, and secondary source. As such, a critical examination and summary of evidence-based knowledge is timely and necessary, so that future trends of host and virus ecology, including seasonal patterns in viral shedding and exposure among wildlife within Arctic regions, can be contextualized based on current consolidated knowledge. This review aims to systematically evaluate the prevalence, spatiotemporal distribution, and ecological characteristics of IAVs detected among wildlife in this understudied region of the globe and compel inquiry into major gaps in knowledge about trends of virus ecology in natural systems which influence the evolution and global dispersal of IAVs.

## 2. Methods

### 2.1. Literature Search and Data Collection

A systematic review of the literature of avian influenza in Arctic regions was performed in accordance with the Preferred Reporting Items for Systematic Reviews and Meta-Analyses (PRISMA) 2020 Guidelines (Appendix A) and registered with the International Prospective Register of Systematic Reviews (PROSPERO; ID CRD42021252595) [34]. The literature search was performed using a set of pre-defined medical subject headings (MeSH) keywords and Boolean operators in PubMed and Google Scholar to identify publications reporting on IAVs in Arctic regions between 1978 and February 2022. The primary keyword “influenza” was used in all searches and was matched with all combinations of secondary and tertiary keywords, specifying geographic locations and ecological factors of interest, respectively. Secondary keywords included: Arctic, Subarctic, circumpolar, polar, Holarctic, Russia, Siberia, Norway, Sweden, Finland, Iceland, Greenland, Canada, Alaska, Svalbard, Bering Sea, and Beringia. Tertiary keywords included: wildlife, avian, bird, animal, marine mammals, seals, whales, sea lion, climate change, warming, ice, permafrost, and environment. All references were exported to EndNote X8.2. Using a two-stage approach, publications were selected for eligibility first based on titles and abstracts, and second based on a full review of article text. 

### 2.2. Inclusion and Exclusion Criteria

To reflect natural infections in wildlife and environmental transmission of IAVs in Arctic regions, all study designs, except experimental laboratory-based studies, and all wildlife species met the criteria for inclusion. Publications from 1978 to February 2022 were investigated for inclusion. Studies predating 1978 were excluded due to the development of Sanger sequencing technology in 1977, which greatly increased the accuracy and robustness of genetic sequencing compared with the years prior, where the assignment of viruses to specific genotypes or subtypes was less accurate and after which data on IAVs became much more accessible [35]. The search was restricted to English language original articles which included the reporting of primary epidemiologic or ecologic data, and review articles that included secondary ecologic data collection and analysis. All publications reporting data from countries with territory in Arctic regions (i.e., Russia, Norway, Sweden, Finland, Iceland, Greenland, Canada, and the United States (specifically Alaska)) were screened, and only those reporting data specifically from regions within the Arctic (defined by the Conservation of Arctic Flora and Fauna’s (CAFF) Arctic Biodiversity Assessment of 2010) were eligible for inclusion [7]. In some cases, articles were included that were not identified through our database searches but later identified upon reviewing the bibliographies of the included publications. Articles containing diagnostic data, incidence, prevalence, seroprevalence, and distribution of IAVs in naturally infected wildlife were included; however, if articles did not report prevalence or seroprevalence, these data were not included in summary tables. In some cases, articles included surveillance activities inside and outside of Arctic regions—if data were not presented separately for Arctic-specific sampling locations, these papers were excluded. Opinion and editorial articles, articles reporting only human infections, and letters to the editor were excluded. Some data were presented in more than one publication, and we removed duplications where relevant. Articles were reviewed independently by two reviewers to determine the final set of included publications in the review. Any discrepancies between the two reviewers were resolved through discussion and consensus.

### 2.3. Data Extraction

An Excel database (v. 16.57) was developed, including the title, authors, year of publication, publication source, and abstract. Where relevant, animal/environmental sources, number of samples analyzed, sample type, diagnostic method, number of positive samples detected, and subtype of detected IAVs were included. The McMaster Critical Review Form for quantitative studies guided our review of the quality and risk of bias in the included studies [36]. Regional data were grouped into distinct geographic Arctic regions for reporting: North Atlantic (combining Iceland and Greenland), Canadian Arctic (data from Canada within CAFF regions), Alaskan Arctic, Russian Arctic (data from Russia within CAFF regions), and Western European Arctic (data associated with Norway, Sweden, and Finland within CAFF regions). 

### 2.4. Statistical Analysis

Prevalence (based on polymerase chain reaction (PCR) results) and seroprevalence (based on blocking enzyme-linked immunosorbent assay (bELISA) results) of IAVs were extracted from included papers and reported by species and year and combined to summarize prevalence and seroprevalence by taxonomic family. A Pearson correlation was performed to determine the association between time in years and number of papers published between 1978 and 2022. The Kruskal–Wallis test was used to determine significant differences in the prevalence of IAVs between (a) Arctic region categories and (b) species categories. Data analyses were performed using JMP Pro v.14.0.0 [37] and SAS software, v.9.4 (SAS Institute, Cary NC, USA) [38].

## 3. Results

### 3.1. Search Results and Study Selection

Our search terms identified 5030 peer-reviewed articles related to IAVs among wildlife or the environment in countries with territory in Arctic regions. During the first selection phase, 2402 (47.8%) duplicates were removed based on PubMed ID, and 503 (10%) were excluded upon title review. Next, the remaining 2125 articles were screened based on review of the title and abstract, with 267 (12.6%) publications being sought for retrieval. Following full-text review, 196 (73.4%) articles did not meet all inclusion criteria, and a total of 71 (26.6%) articles were included in the full review. No studies were excluded for issues of quality or bias (Figure 2, Appendix A). 

### 3.2. Characteristics of the Included Studies

Reviewed articles eligible for inclusion were published between 1993 and 2022, from all countries with territory in the Arctic region. A total of 5 (7%) articles were published between 1993 and 2000, 18 (25%) between 2001 and 2010, and 48 (68%) between 2011 and 2022. There is a positive correlation between the year of publication and the number of articles published (r = 0.653, 95% CI [0.339, 0.836], *p* = 0.0005), demonstrating a significant increase in the number of published articles between the years 1993 and 2022 (Figure 3).

### 3.3. Natural History and Epidemiology of IAVs in Arctic Regions

IAVs have been detected in multiple wildlife species in Arctic regions, including seabirds, shorebirds, waterfowl, seals, sea lions, whales, and terrestrial mammals, and in the environment [11,39,40,41,42,43,44,45]. IAV genomic sequence data are available through multiple publicly accessible databases (i.e., GenBank, NCBI, IVR, GISAID); however, many published sequences from countries with territory in Arctic regions lack specific sampling location data, making it difficult to identify sequences isolated specifically from these regions. Additionally, many surveillance activities result in IAV genomic sequences that are never reported in the peer-review literature; therefore, relying solely on publicly available sequence counts misrepresents the true proportion of virus detection among wildlife in Arctic regions. Prevalence and seroprevalence proportions reported in the literature provide a comprehensive overview of the natural history of IAVs in Arctic regions. Seroprevalence proportions, in particular, provide data on IAV exposure throughout the duration of wildlife hosts’ immunological memory, suggesting that low prevalence rates may be linked to the timing of surveillance activities and/or short periods of active viral shedding [46].

Herein, we present peer-reviewed data on prevalence and seroprevalence among wild birds, mammals, and environmental matrices sampled in Arctic regions between 1978 and February 2022.

### 3.4. A Broad Avian Host Range for Numerous Subtypes of IAVs in Arctic Regions

Wild aquatic avian species, specifically *Anseriformes* (waterfowl, including ducks, geese, and swans) and *Charadriiformes* (seabirds, including gulls, shorebirds, terns, and auks), comprise the native host reservoir for IAVs worldwide, and likewise in Arctic regions where a broad diversity of seabirds, shorebirds, and waterfowl predominate [16,47]. Wild avian hosts of IAVs sampled in Arctic regions include resident birds that live in the region year-round (i.e., Sabine’s Gull, Rock Ptarmigan) and temporary Arctic residents (i.e., Mallards, Northern Wheatear, Great Black-backed Gull, Common Murre) that migrate to Arctic regions in the spring and summer months to breed and nest, taking advantage of the region’s abundant energy resources [39,42,48,49,50,51].

Peer-reviewed reports of IAVs in avian species were collected from all geographic Arctic regions, including the North Atlantic (Iceland (*n* = 2) and Greenland (*n* = 3)), Western European Arctic (Sweden (*n* = 0), Finland (*n* = 0), and Norway (*n* = 1)), Russian Arctic (*n* = 2), Canadian Arctic (*n* = 7), and Alaskan Arctic (*n* = 17) (Table 1). Studies implemented a range of detection methods including PCR and serology. In most serological studies, serum samples were screened for antibodies to IAVs using commercially available blocking enzyme-linked immunosorbent assay (bELISA) targeting the IAV nucleoprotein. Some studies characterized subtype-specific antibodies using hemagglutination inhibition (HI) assays.

A total of 43,368 individual birds were sampled in Arctic regions, representing a broad diversity of 171 species, with the most samples collected from Anatidae, a family of water birds including ducks, geese, and swans. When grouped by avian species categories, studies demonstrated a total pooled prevalence of 4.23% among all host species, and 5.84% among Anatidae, 2.12% among Laridae, and 1.51% among all other species combined (Figure 4). Pooled prevalence among wild birds by Arctic regions included 0.69% in the North Atlantic, 13.95% in the Western European Arctic, 0.21% in the Russian Arctic, 3.46% in the Canadian Arctic, and 5.02% in Alaska, USA. There were not significant differences in prevalence between Anatidae, Laridae, and all other avian species (*p* = 0.3679) or Arctic regions (*p* = 0.4060). Sample sizes varied considerably between studies and species. Subtypes generally clustered by host species, with H13, H16, H1, and H9 subtypes commonly identified in gull species and H3, H4, and H5 more commonly found in duck species. Host-specificity of H13 and H16 subtypes has been commonly reported for gulls, whereas ducks are considered to be more permissive hosts to a wider range of subtypes [52]. In total, 12 unique hemagglutinin-neuraminidase (HANA) subtypes have been detected from the literature on wild birds in Arctic regions (Table 1).

Seroprevalence rates were generally much higher than viral shedding, reflecting exposure throughout the lifecycle. While a few studies detected no antibodies during serological screening, seroprevalence of IAV antibodies ranged from 3.4% in Yellow-billed Loons of the Beaufort Sea coast, Alaska (2012–2016) to 58.2% in Black-legged Kittiwakes in Hornøya, Norway (2008–2009) [53,54]. Subtype-specific antibodies were reported in a small subset of studies, including H4N2 (5% (1/20) among Red-throated Loons) and H1N2 (26.6% (4/15) among greater white-fronted geese) in Alaska [40]. High subtype-specific seroprevalence of H13 (18/48 (37.5%)) and H16 (39/48 (81.3%)) antibodies were detected in Black-legged Kittiwakes in Arctic Norway, which reflects host-specificity of these gull-adapted subtypes [53,55]. Additionally, sera from Emperor Geese in Alaska (2015–2017) were shown to be reactive to a large diversity of HA subtypes, suggesting these hosts may play a substantial role in the maintenance and spread of IAVs in Alaska and throughout the Beringia region [56] (Table 1).

**Table 1 viruses-14-01531-t001:** Prevalence and seroprevalence rates of IAVs among avian species in Arctic regions.

Family Taxa	Host Species	Sampling Year(s)	Location	Prevalence				Seroprevalence		Author, Year
				Positive Samples	Total Samples	Prevalence %	Virus Subtypes Identified (LP: Low Path, HP: High Path)	*n*/*N* (%)	Subtype-Specific Antibodies Identified, *n*/*N* (%) (LP: Low Path, HP: High Path)	
**Accipitridae**				**0**	**0**	**0%**				
	Northern Goshawk (*Accipiter gentilis*)	2016	Troms, Norway	0	-	0%	-	0/29 (0)	-	Lee, 2019 [41]
	White-tailed Eagle (*Haliaeetus albicilla*)	2016	Steigen, Norway	0	-	0%	-	0/43 (0)	-	Lee, 2019 [41]
**Alcidae**				**150**	**3685**	**4%**				
	Dovekie (*Alle alle*)	2008–2011	Newfoundland and Labrador, Canada	0	52	0%	-	-	-	Wille, 2014 [57]
	Crested Auklet (*Aethia cristatetlla*)	2006–2007	Alaska	0	207	0%	-	-	-	Ip, 2008 [58]
	Parakeet Auklet (*Aethia psittacula*)	2006–2007	Alaska	0	8	0%	-	-	-	Ip, 2008 [58]
	Least Auklet (*Aethia pusilla*)	2006–2007	Alaska	0	30	0%	-	-	-	Ip, 2008 [58]
	Razorbill (*Alca torda*)	2008–2011	Newfoundland and Labrador, Canada	0	196	0%	-	-	-	Wille, 2014 [57]
	Black Guillemot (*Cepphus grille*)	2008–2011	Newfoundland and Labrador, Canada	0	1	0%	-	-	-	Wille, 2014 [57]
		2010–2011	Iceland	0	10	0%	-	-	-	Dusek, 2014 [4]
	Atlantic Puffin (*Fratercula arctica*)	2008–2011	Newfoundland and Labrador, Canada	0	365	0%	-	-	-	Wille, 2014 [57]
	Tufted Puffin (*Fratercula cirrhata*)	2006–2007	Alaska	0	4	0%	-	-	-	Ip, 2008 [58]
	Horned Puffin (*Fratercula corniculata*)	2006–2007	Alaska	0	2	0%	-	-	-	Ip, 2008 [58]
	Common Murre (*Uria aalge*)	2006–2007	Alaska	1	76	1%	-	-	-	Ip, 2008 [58]
		2008–2011	Newfoundland and Labrador, Canada	70	1273	6%	-	-	-	Wille, 2014 [57]
		2010–2011	Iceland	0	1	0%	-	-	-	Dusek, 2014 [4]
		2011	Newfoundland and Labrador, Canada	68	452	15%	-	-	-	Huang, 2014 [59]
	Thick-billed Murre (*Uria lomvia*)	2006–2007	Alaska	7	235	3%	-	-	-	Ip, 2008 [58]
		2007–2011	Greenland	0	64	0%	-	-	-	Hjulsager, 2012 [42]
		2008–2011	Newfoundland and Labrador, Canada	1	621	0%	-	-	-	Wille, 2014 [57]
		2014	Greenland	2	44	5%	H11N2: 2 birds	-	-	Hartby, 2016 [60]
		2014	Greenland	1	44	2%	H5N1 (LP): 1 bird	-	-	Hartby, 2016 [60]
**Aluadidae**				**0**	**1**	**0%**				
	Horned Lark (*Eremophila alpestris*)	2006–2007	Alaska	0	1	0%	-	-	-	Ip, 2008 [58]
**Anatidae Anatini**				**1142**	**10,022**	**11%**				
	Northern Pintail (*Anas acuta*)	2005	Minto Flats State Game Refuge, Alaska	1	19	5%	-	-	-	Runstadler, 2007 [61]
		2005	Minto Flats State Game Refuge, Alaska	178	738	24%	H3N6: 3/38 (7.9%); H4N6: 9/38 (23.7%); H8N4: 1/38 (2.6%); H12N5: 7/38 (18.4%)	-	-	Runstadler, 2007 [61]
		2005	Pt. Barrow, Alaska	3	133	2%	-	-	-	Wahlgren, 2008 [62]
		2005	Chuchki Peninsula, Russia	0	2	0%	-	-	-	Wahlgren, 2008 [62]
		2006–2007	Alaska	51	539	10%	-	-	-	Ip, 2008 [58]
		2006–2007	Alaska	87	1426	6%	-	-	-	Ip, 2008 [58]
		2006–2008	Izembek Lagoon, Alaska	29	258	11%	-	-	-	Ramey, 2011 [63]
		2007–2008	Minto Flats State Game Refuge, Alaska	101	1291	8%	-	-	-	Hill, 2012 [3]
		2007–2008	Minto Flats State Game Refuge, Alaska	19	215	9%	-	-	-	Hill, 2012 [64]
		2011–2013	Izembek National Wildlife Refuge, Alaska	104	709	15%	-	-	-	Ramey, 2015 [65]; Reeves, 2018 [66]
		2014	Izembek National Wildlife Refuge, Alaska	17	69	25%	-	-	-	Ramey, 2016 [67]; Reeves, 2018 [66]
		2015	Alaska	-	-	-	-	28/209 (13.4%)	H5 *: 0/45 (0%)	Stallknecht, 2020 [49]
		2015	Yukon-Kuskokwim Delta, Alaska	1	15	7%	-	-	-	Ramey, 2016 [67]
	American Wigeon (*Anas americana*)	2005	Minto Flats State Game Refuge, Alaska	1	2	50%	-	-	-	Runstadler, 2007 [61]
		2006–2007	Alaska	0	1	0%	-	-	-	Ip, 2008 [58]
		2007–2008	Minto Flats State Game Refuge, Alaska	7	241	3%	-	-	-	Hill, 2012 [64]
		2011–2013	Izembek National Wildlife Refuge, Alaska	0	34	0%	-	-	-	Ramey, 2015 [65]; Reeves, 2018 [66]
		2014	Izembek National Wildlife Refuge, Alaska	1	11	9%	-	-	-	Ramey, 2016 [67]; Reeves, 2018 [66]
		2015	Yukon-Kuskokwim Delta, Alaska	0	1	0%	-	-	-	Ramey, 2016 [67]
	Northern Shoveler (*Anas clypeata*)	2006–2007	Alaska	1	22	5%	-	-	-	Ip, 2008 [58]
		2007–2008	Minto Flats State Game Refuge, Alaska	38	118	32%	-	-	-	Hill, 2012 [64]
		2011–2013	Izembek National Wildlife Refuge, Alaska	1	5	20%	-	-	-	Ramey, 2015 [65]; Reeves, 2018 [66]
		2014	Izembek National Wildlife Refuge, Alaska	1	5	20%	-	-	-	Ramey, 2016 [68]; Reeves, 2018 [66]
		2015	Yukon-Kuskokwim Delta, Alaska	1	19	5%	-	-	-	Ramey, 2016 [67]
	American Green-winged Teal (*Anas crecca*)	2011–2013	Izembek National Wildlife Refuge, Alaska	26	111	23%	-	-	-	Ramey, 2015 [69]; Reeves, 2018 [66]
		2015	Yukon-Kuskokwim Delta, Alaska	0	4	0%	-	-	-	Ramey, 2016 [67]
	Blue-winged Teal (*Anas discors*)	2006–2007	Alaska	0	1	0%	-	-	-	Ip, 2008 [58]
	Eurasian Wigeon (*Anas Penelope*)	2006–2007	Alaska	0	1	0%	-	-	-	Ip, 2008 [58]
		2011–2013	Izembek National Wildlife Refuge, Alaska	0	18	0%	-	-	-	Ramey, 2015 [69]
		2014	Belaya Gora, Sakha Republic, Russia	1			H5N8 (HP) (tracheal specimen)	-	-	Marchenko, 2015 [70]
		2014	Izembek National Wildlife Refuge, Alaska	0	1	0%	-	-	-	Ramey, 2016 [68]
	Mallard (*Anas platyrhynchos*)	2005	Minto Flats State Game Refuge, Alaska	45	121	37%	H3N8: 2/38 (5.2%); H4N6: 2/38 (5.2%); H8N4: 2/38 (5.2%); H12N5: 2/38 (5.2%)	-	-	Runstadler, 2007 [61]
		2006–2007	Alaska	25	273	9%	-	-	-	Ip, 2008 [58]
		2007–2011	Greenland	2	623	0%	-	-	-	Hjulsager, 2012 [42]
		2007–2008	Minto Flats State Game Refuge, Alaska	124	1004	12%	-	-	-	Hill, 2012 [3]
		2010–2011	Iceland	1	14	7%	-	-	-	Dusek, 2014 [4]
		2011–2013	Izembek National Wildlife Refuge, Alaska	7	72	10%	-	-	-	Ramey, 2015 [69]; Reeves, 2018 [66]
		2012–2015	Fairbanks and Anchorage, Alaska	181	1062	17%	-	457/984 (46.4%)	-	Spivey, 2017 [71]
		2013–2014	Subarctic Ontario, Canada	12	304	4%	H3: 3/12 (25%) H4: 1/12 (8.3%)	-	-	Liberda, 2017 [48]
		2014	Izembek National Wildlife Refuge, Alaska	18	49	37%	-	-	-	Ramey, 2016 [68]; Reeves, 2018 [66]
		2014–2015	Anchorage, Alaska	63	700		-	-	-	Hill, 2017 [10]
		2015	Alaska	-			-	98/210 (46.7%)	H5 *: 24/91 (26.4%)	Stallknecht, 2020 [49]
		2015	Yukon-Kuskokwim Delta, Alaska	0	12	0%	-	-	-	Ramey, 2016 [67]
	Gadwall (*Anas strepera*)	2006–2007	Alaska	0	16	0%	-	-	-	Ip, 2008 [58]
		2007–2008	Minto Flats State Game Refuge, Alaska	0	1	0%	-	-	-	Hill, 2012 [3]
		2011–2013	Izembek National Wildlife Refuge, Alaska	0	14	0%	-	-	-	Ramey, 2015 [69]
		2015	Yukon-Kuskokwim Delta, Alaska	0	1	0%	-	-	-	Ramey, 2016 [67]
	Miscellaneous duck species ***	2005	Pt. Barrow, Alaska	0	1	0%	-	-	-	Wahlgren, 2008 [62]
		2005	Newfoundland and Labrador, Canada	8	73	11%	-	-	-	Parmley, 2008 [72]
		2013–2014	St. Paul Island and Prince William Sound, Alaska	2	185	1%	-	-	-	Ramey, 2015 [65]
		2016	Fairbanks, Alaska	48	188	26%	H5N2 (HP): 1/188 (0.5%)	-	-	Lee, 2017 [73]
**Anatidae Anserini**				**322**	**11,636**	**3%**				
	Greater White-fronted Goose (*Anser albifrons*)	2005	Pt. Barrow, Alaska	0	11	0%	-	-	-	Wahlgren, 2008 [62]
		2006–2007	Alaska	40	1081	4%	-	-	-	Ip, 2008 [58]
		2007–2008	Minto Flats State Game Refuge, Alaska	0	3	0%	-	-	-	Hill, 2012 [3]
		2007–2011	Greenland	0	179	0%	-	-	-	Hjulsager, 2012 [42]
		2011–2013	Izembek National Wildlife Refuge, Alaska	0	2	0%	-	-	-	Ramey, 2015 [69]
		2015	Yukon-Kuskokwim Delta, Alaska	39	298	13%	-	-	-	Ramey, 2016 [67]
		2012–2016	Alaska	-			-	35/76 (46%)	H1N2: 4/15 (26.6%)	Van Hemert, 2019 [40]
									H6N2: 5/15 (33.3%)	
									H9N2: 6/15 (40%)	
	Greylag Goose (*Anser anser*)	2010–2011	Iceland	11	223	5%	-	-	-	Dusek, 2014 [4]
	Pink-footed Goose (*Anser brachyrhynchus*)	2010–2011	Iceland	1	13	8%	-	-	-	Dusek, 2014 [4]
		2016	Greenland	0	150	0%	-	-	-	Gaidet, 2018 [74]
	Brent Goose (*Branta bernicula*)	2005	Chukchi Peninsula, Russia	0	12	0%	-	-	-	Wahlgren, 2008 [62]
		2005	Pt. Barrow, Alaska	0	1	0%	-	-	-	Wahlgren, 2008 [62]
	Black Brant (*Branta bernicla nigricans*)	2006–2007	Alaska	8	2075	0%	-	-	-	Ip, 2008 [58]
		2011–2013	Izembek National Wildlife Refuge, Alaska	0	192	0%	-	-	-	Ramey, 2015 [69]; Reeves, 2018 [66]
		2012–2016	Alaska	-			-	56/110 (50.9%)	-	Van Hemert, 2019 [40]
		2014	Izembek National Wildlife Refuge, Alaska	0	21	0%	-	-	-	Ramey, 2016 [68]; Reeves, 2018 [66]
		2015	Yukon-Kuskokwim Delta, Alaska	3	41	7%	-	-	-	Ramey, 2016 [67]
	Canada Goose (*Branta canadensis*)	2006–2007	Alaska	9	260	4%	-	-	-	Ip, 2008 [58]
		2007–2011	Greenland	0	221	0%	-	-	-	Hjulsager, 2012 [42]
	Cackling Goose (*Branta hutchinsii*)	2006–2007	Alaska	2	373	1%	-	-	-	Ip, 2008 [58]
		2011–2013	Izembek National Wildlife Refuge, Alaska	5	221	2%	-	-	-	Ramey, 2015 [69]; Reeves, 2018 [66]
		2015	Alaska	-			-	33/105 (31.4%)	H5 *: 1/23 (4.3%)	Stallknecht, 2020 [49]
		2015	Yukon-Kuskokwim Delta, Alaska	9	315	3%	-	-	-	Ramey, 2016 [67]
	Barnacle Goose (*Branta leucopsis*)	2016	Greenland	0	150	0%	-	-	-	Gaidet, 2018 [74]
	Emperor Goose (*Chen canagica*)	2005	Chukchi Peninsula, Russia	0	19	0%	-	-	-	Wahlgren, 2008 [62]
		2006–2007	Alaska	11	685	2%	-	-	-	Ip, 2008 [58]
		2011–2013	Izembek National Wildlife Refuge, Alaska	26	663	4%	-	-	-	Ramey, 2015 [69]
		2014	Izembek National Wildlife Refuge, Alaska	14	294	5%	-	-	H1–H12, H14, H15 **	Ramey, 2016 [68]; Reeves, 2018 [66]
		2015	Yukon-Kuskokwim Delta, Alaska	4	19	21%	-	-	-	Ramey, 2016 [67]
		2015–2016	Izembek National Wildlife Refuge, Alaska	59	390	15%	-	-	H1–H12, H14, H15 **	Ramey, 2019 [56]
		2015–2017	Kodiak Island, Alaska	41	800	5%	-	-	H1–H12, H14, H15 **	Ramey, 2019 [56]
		2016	Shemya Island, Alaska	2	400	1%	-	-	H1–H12, H14, H15 **	Ramey, 2019 [56]
		2016	Nelson Lagoon, Alaska	1	194	1%	-	-	H1–H12, H14, H15 **	Ramey, 2019 [56]
		2016	Cinder Lagoon, Alaska	3	225	1%	-	-	H1–H12, H14, H15 **	Ramey, 2019 [56]
		2016–2017	Adak Island, Alaska	9	800	1%	-	-	H1–H12, H14, H15 **	Ramey, 2019 [56]
	Snow Goose (*Chen caerulescens*)	1993–1996	Wrangel Island, Russia	-	-	-	-	513/983 (52.2%)	-	Samuel, 2015 [75]
		1993–1996	Banks Island, Canada	-	-	-	-	855/1996 (42.8%)	-	Samuel, 2015 [75]
		2005	Chukchi Peninsula, Russia	0	222	0%	-	-	-	Wahlgren, 2008 [62]
		2006–2007	Alaska	17	720	2%	-	-	-	Ip, 2008 [58]
		2007–2011	Greenland	0	283	0%	-	-	-	Hjulsager, 2012 [42]
		2007–2008	Minto Flats State Game Refuge, Alaska	0	2	0%	-	-	-	Hill, 2012 [3]
		2013–2014	Subarctic Ontario, Canada	3	16	19%	-	-	-	Liberda, 2017 [48]
		2015	Alaska	-			-	92/200 (46%)	H5 *: 8/75 (10.7%)	Stallknecht, 2020 [49]
		2015	Yukon-Kuskokwim Delta, Alaska	5	62	8%	-	-	-	Ramey, 2016 [67]
		2012–2016	Alaska	-			-	57/108 (52.8%)	-	Van Hemert, 2019 [40]
**Anatidae Aythyini**				**10**	**221**	**5%**				
	Redhead (*Aythya americana*)	2006–2007	Alaska	0	1	0%	-	-	-	Ip, 2008 [58]
	Lesser Scaup (*Aythya affinis*)	2006–2007	Alaska	0	19	0%	-	-	-	Ip, 2008 [58]
		2007–2008	Minto Flats State Game Refuge, Alaska	0	19	0%	-	-	-	Hill, 2012 [3]
		2011–2013	Izembek National Wildlife Refuge, Alaska	0	2	0%	-	-	-	Ramey, 2015 [69]
		2015	Yukon-Kuskokwim Delta, Alaska	7	62	11%	-	-	-	Ramey, 2016 [67]
	Ring-necked Duck (*Aythya collaris*)	2007–2008	Minto Flats State Game Refuge, Alaska	0	16	0%	-	-	-	Hill, 2012 [3]
		2011–2013	Izembek National Wildlife Refuge, Alaska	0	1	0%	-	-	-	Ramey, 2015 [69]
	Canvasback (*Aythya valisineria*)	2006–2007	Alaska	0	8	0%	-	-	-	Ip, 2008 [58]
		2007–2008	Minto Flats State Game Refuge, Alaska	0	23	0%	-	-	-	Hill, 2012 [3]
		2014	Izembek National Wildlife Refuge, Alaska	0	1	0%	-	-	-	Ramey, 2016 [68]
	Greater Scaup (*Aythya marila*)	2006–2007	Alaska	0	1	0%	-	-	-	Ip, 2008 [58]
		2011–2013	Izembek National Wildlife Refuge, Alaska	1	36	3%	-	-	-	Ramey, 2015 [69]; Reeves, 2018 [66]
		2014	Izembek National Wildlife Refuge, Alaska	0	21	0%	-	-	-	Ramey, 2016 [68]; Reeves, 2018 [66]
		2015	Yukon-Kuskokwim Delta, Alaska	2	11	18%	-	-	-	Ramey, 2016 [67]
**Anatidae Cygnini**				**10**	**637**	**2%**				
	Trumpeter Swan (*Cyngus buccinator*)	2006–2007	Alaska	0	10	0%	-	-	-	Ip, 2008 [58]
	Tundra Swan (*Cyngus columbianus*)	2005	Chukchi Peninsula, Russia	0	3	0%	-	-	-	Wahlgren, 2008 [62]
		2006–2007	Alaska	7	583	1%	-	-	-	Ip, 2008 [58]
		2015	Yukon-Kuskokwim Delta, Alaska	3	41	7%	-	-	-	Ramey, 2016 [67]
**Anatidae Mergini**				**62**	**3956**	**2%**				
	Bufflehead (*Bucephala albeola*)	2006–2007	Alaska	0	1	0%	-	-	-	Ip, 2008 [58]
		2007–2008	Minto Flats State Game Refuge, Alaska	0	29	0%	-	-	-	Hill, 2012 [3]
		2011–2013	Izembek National Wildlife Refuge, Alaska	3	15	20%	-	-	-	Ramey, 2015 [69]; Reeves, 2018 [66]
		2014	Izembek National Wildlife Refuge, Alaska	2	10	20%	-	-	-	Ramey, 2016 [68]; Reeves, 2018 [66]
	Common Goldeneye (*Bucephala clangula*)	2011–2013	Izembek National Wildlife Refuge, Alaska	0	2	0%	-	-	-	Ramey, 2015 [69]
		2014	Izembek National Wildlife Refuge, Alaska	0	2	0%	-	-	-	Ramey, 2016 [68]
		2015	Yukon-Kuskokwim Delta, Alaska	0	2	0%	-	-	-	Ramey, 2016 [67]
	Long-tailed Duck (*Clangula hyemalis*)	2006–2007	Alaska	1	52	2%	-	-	-	Ip, 2008 [58]
		2011–2013	Izembek National Wildlife Refuge, Alaska	1	16	6%	-	-	-	Ramey, 2015 [69]; Reeves, 2018 [66]
		2015	Yukon-Kuskokwim Delta, Alaska	1	6	17%	-	-	-	Ramey, 2016 [67]
	Harlequin Duck (*Histronicus histronicus*)	2006–2007	Alaska	0	65	0%	-	-	-	Ip, 2008 [58]
		2011–2013	Izembek National Wildlife Refuge, Alaska	0	48	0%	-	-	-	Ramey, 2015 [69]
		2014	Izembek National Wildlife Refuge, Alaska	0	2	0%	-	-	-	Ramey, 2016 [68]
	White-winged Scoter (*Malanitta fusca*)	2006–2007	Alaska	0	41	0%	-	-	-	Ip, 2008 [58]
		2011–2013	Izembek National Wildlife Refuge, Alaska	0	15	0%	-	-	-	Ramey, 2015 [69]
		2014	Izembek National Wildlife Refuge, Alaska	0	23	0%	-	-	-	Ramey, 2016 [68]
		2015	Yukon-Kuskokwim Delta, Alaska	1	16	6%	-	-	-	Ramey, 2016 [67]
	Black Scoter (*Melanitta nigra*)	2006–2007	Alaska	0	10	0%	-	-	-	Ip, 2008 [58]
		2011–2013	Izembek National Wildlife Refuge, Alaska	0	9	0%	-	-	-	Ramey, 2015 [69]
		2014	Izembek National Wildlife Refuge, Alaska	0	1	0%	-	-	-	Ramey, 2016 [68]
		2015	Yukon-Kuskokwim Delta, Alaska	4	83	5%	-	-	-	Ramey, 2016 [67]
	Surf Scoter (*Melanitta perspicillata*)	2006–2007	Alaska	0	2	0%	-	-	-	Ip, 2008 [58]
		2015	Yukon-Kuskokwim Delta, Alaska	1	5	20%	-	-	-	Ramey, 2016 [67]
	Common Merganser (*Mergus merganser*)	2006–2007	Alaska	0	1	0%	-	-	-	Ip, 2008 [58]
		2015	Yukon-Kuskokwim Delta, Alaska	0	1	0%	-	-	-	Ramey, 2016 [67]
	Red-breasted Merganser (*Mergus serrator*)	2007–2011	Greenland	0	5	0%	-	-	-	Hjulsager, 2012 [42]
		2011–2013	Izembek National Wildlife Refuge, Alaska	0	4	0%	-	-	-	Ramey, 2015 [69]
		2014	Izembek National Wildlife Refuge, Alaska	0	6	0%	-	-	-	Ramey, 2016 [68]
	Steller’s Eider (*Polysticta stelleri*)	2006–2007	Alaska	6	737	1%	-	-	-	Ip, 2008 [58]
		2006–2008	Izembek Lagoon, Alaska	1	457	0%	-	-	-	Ramey, 2011 [63]
		2006–2008	Nelson Lagoon, Alaska	30	779	4%	-	-	-	Ramey, 2011 [63]
	Spectacled Eider (*Somateria fischeri*)	2006–2007	Alaska	2	348	1%	-	-	-	Ip, 2008 [58]
	Common Eider (*Somateria mollissima*)	2005	Chukchi Peninsula, Russia	0	7	0%	-	-	-	Wahlgren, 2008 [62]
		2006–2007	Alaska	1	395	0%	-	-	-	Ip, 2008 [58]
		2007–2011	Greenland	0	20	0%	-	-	-	Hjulsager, 2012 [42]
		2007–2011	Nunavut, Canada	-			-	304/552 (55.1%)	-	Hall, 2015 [76]
		2010–2011	Iceland	0	35	0%	-	-	-	Dusek, 2014 [4]
		2011–2013	Izembek National Wildlife Refuge, Alaska	0	20	0%	-	-	-	Ramey, 2015 [69]
		2012	Iceland	-			-	31/38 (81.6%)	-	Hall, 2015 [76]
		2014	Izembek National Wildlife Refuge, Alaska	0	1	0%	-	-	-	Ramey, 2016 [68]
	King Eider (*Somateria spectabilis*)	2006–2007	Alaska	6	680	1%	-	-	-	Ip, 2008 [58]
		2011–2013	Izembek National Wildlife Refuge, Alaska	2	5	40%	-	-	-	Ramey, 2015 [69]; Reeves, 2018 [66]
**Certhiidae**				**0**	**1**	**0%**				
	Brown Creeper (*Certhia americana*)	2006–2007	Alaska	0	1	0%	-	-	-	Ip, 2008 [58]
**Charadriidae**				**0**	**177**	**0%**				
	Common Ringed Plover (*Charadrius hiaticula*)	2005	Chukchi Peninsula, Russia	0	10	0%	-	-	-	Wahlgren, 2008 [62]
		2010–2011	Iceland	0	4	0%	-	-	-	Dusek, 2014 [4]
		2012	Iceland	0	10	0%	-	0/10 (0%)	-	Hall, 2014 [77]
		2013	Iceland	0	13	0%	-	-	-	Hall, 2014 [77]
	Semipalmated Plover (*Charadrius semipalmatus*)	2006–2007	Alaska	0	4	0%	-	-	-	Ip, 2008 [58]
	American Golden Plover (*Pluvialis dominica*)	2006–2007	Alaska	0	15	0%	-	-	-	Ip, 2008 [58]
	Pacific Golden Plover (*Pluvialis fulva*)	2006–2007	Alaska	0	42	0%	-	-	-	Ip, 2008 [58]
	Black-bellied Plover (*Pluvialis squatarola*)	2006–2007	Alaska	0	73	0%	-	-	-	Ip, 2008 [58]
	Golden Plover spp. (*Pluvialis* spp.)	2006–2007	Alaska	0	6	0%	-	-	-	Ip, 2008 [58]
**Corvidae**				**0**	**201**	**0%**				
	Common Raven (*Corus corax*)	2007–2011	Greenland	0	199	0%	-	-	-	Hjulsager, 2012 [42]
	Gray Jay (*Perisoreus canadensis*)	2006–2007	Alaska	0	2	0%	-	-	-	Ip, 2008 [58]
**Emberizidae**				**0**	**463**	**0%**				
	Lapland Longspur (*Calcarius lapponicus*)	2006–2007	Alaska	0	8	0%	-	-	-	Ip, 2008 [58]
		2007–2011	Greenland	0	51	0%	-	-	-	Hjulsager, 2012 [42]
		2015	Yukon-Kuskokwim Delta, Alaska	0	1	0%	-	-	-	Ramey, 2016 [67]
		2005	Chukchi Peninsula, Russia	0	2	0%	-	-	-	Wahlgren, 2008 [62]
	Dark-eyed Junco (*Junco hyemalis*)	2006–2007	Alaska	0	16	0%	-	-	-	Ip, 2008 [58]
	Lincoln’s Sparrow (*Melospiza lincolnii*)	2006–2007	Alaska	0	20	0%	-	-	-	Ip, 2008 [58]
	Fox Sparrow (*Passerella iliaca*)	2006–2007	Alaska	0	26	0%	-	-	-	Ip, 2008 [58]
	Savannah Sparrow (*Passerculus sandwichensis*)	2006–2007	Alaska	0	92	0%	-	-	-	Ip, 2008 [58]
	Snow Bunting (*Plectrophenax nivalis*)	2006–2007	Alaska	0	2	0%	-	-	-	Ip, 2008 [58]
		2007–2011	Greenland	0	80	0%	-	-	-	Hjulsager, 2012 [42]
	American Tree Sparrow (*Spizella arborea*)	2006–2007	Alaska	0	96	0%	-	-	-	Ip, 2008 [58]
	Golden-crowned Sparrow (*Zonotrichia atricapilla*)	2006–2007	Alaska	0	20	0%	-	-	-	Ip, 2008 [58]
	White-crowned Sparrow (*Zonotrichia leucophrys*)	2006–2007	Alaska	0	49	0%	-	-	-	Ip, 2008 [58]
**Falconidae**				**0**	**17**	**0%**				
	Peregrine Falcon (*Falco peregrinus*)	2007–2011	Greenland	0	16	0%	-	-	-	Hjulsager, 2012 [42]
	Greyfalcon (*Falco rusticolus*)	2014	Izembek National Wildlife Refuge, Alaska	0	1	0%	-	-	-	Ramey, 2016 [68]
**Fringillidae**				**0**	**149**	**0%**				
	Common Redpoll (*Acanthis flammea*)	2007–2011	Greenland	0	82	0%	-	-	-	Hjulsager, 2012 [42]
	Hoary Redpoll (*Carduelis hornemanni*)	2006–2007	Alaska	0	10	0%	-	-	-	Ip, 2008 [58]
	Common Redpoll (*Carduelis flammea*)	2006–2007	Alaska	0	52	0%	-	-	-	Ip, 2008 [58]
	Brambling (*Fringilla montifringilla*)	2006–2007	Alaska	0	1	0%	-	-	-	Ip, 2008 [58]
	White-winged Crossbill (*Loxia leucoptera*)	2006–2007	Alaska	0	1	0%	-	-	-	Ip, 2008 [58]
	Pine Grosbeak (*Pinicola enucleator*)	2006–2007	Alaska	0	3	0%	-	-	-	Ip, 2008 [58]
**Gaviidae**				**1**	**10**	**10%**				
	Yellow-billed Loon (*Gavia adamsii*)	2006–2007	Alaska	0	2	0%	-	-	-	Ip, 2008 [58]
		2012–2014	Beaufort Sea coast, Alaska	-			-	1/29 (3.4%)	-	Uher-Koch, 2019 [54]
		2012–2016	Alaska	-			-	1/29 (3.4%)	H5N2: 1/20 (5%)	Van Hemert, 2019 [40]
		2017	Chukchi Sea coast, Alaska	-			-	1/7 (14.2%)	-	Uher-Koch, 2019 [54]
	Arctic Loon (*Gavia arctica*)	2006–2007	Alaska	0	1	0%	-	-	-	Ip, 2008 [58]
	Common Loon (*Gavia immer*)	2006–2007	Alaska	0	1	0%	-	-	-	Ip, 2008 [58]
	Pacific Loon (*Gavia pacifica*)	2006–2007	Alaska	0	2	0%	-	-	-	Ip, 2008 [58]
		2008–2010	Chukchi Sea coast, Alaska	-			-	15/28 (53.6%)	-	Uher-Koch, 2019 [54]
		2012–2016	Alaska	-			-	5/48 (10.4%)	-	Van Hemert, 2019 [40]
		2012–2016	Beaufort Sea coast, Alaska	-			-	5/48 (10.4 %)	-	Uher-Koch, 2019 [54]
		2015	Yukon-Kuskokwim Delta, Alaska	1	1	100%	-	-	-	Ramey, 2016 [67]
		2016	Yukon-Kuskokwim Delta, Alaska	-			-	2/14 (14.2%)	-	Uher-Koch, 2019 [54]
	Red-throated Loon (*Gavia stellata*)	2006–2007	Alaska	0	3	0%	-	-	-	Ip, 2008 [58]
		2008–2010	Chukchi Sea coast, Alaska	-			-	15/33 (45.5%)	-	Uher-Koch, 2019 [54]
		2010, 2012–2014	Beaufort Sea coast, Alaska	-			-	3/13 (23%)	-	Uher-Koch, 2019 [54]
		2012–2016	Alaska	-			-	2/10 (20%)	H4N2: 1/20 (5%)	Van Hemert, 2019 [40]
**Gruidae**				**2**	**192**	**1%**				
	Sandhill Crane (*Grus canadensis*)	2006–2007	Alaska	0	161	0%	-	-	-	Ip, 2008 [58]
		2015	Yukon-Kuskokwim Delta, Alaska	2	31	7%	-	-	-	Ramey, 2016 [67]
**Haematopodidae**				**0**	**16**	**0%**				
	Eurasian Oystercatcher (*Haematopus ostralegus*)	2010–2011	Iceland	0	16	0%	-	-	-	Dusek, 2014 [4]
**Hydrobatidae**				**0**	**377**	**0%**				
	Leach’s Storm Petrel (*Oceandroma leucorhoa*)	2008–2011	Newfoundland and Labrador, Canada	0	377	0%	-	-	-	Wille, 2014 [57]
**Hirundinidae**				**0**	**2**	**0%**				
	Cliff Swallow (*Petrochelidon pyrrhonota*)	2006–2007	Alaska	0	1	0%	-	-	-	Ip, 2008 [58]
	Tree Swallow (*Tachycineta bicolor*)	2006–2007	Alaska	0	1	0%	-	-	-	Ip, 2008 [58]
**Laniida**				**0**	**2**	**0%**				
	Northern Shrike (*Lanius excubitor*)	2006–2007	Alaska	0	2	0%	-	-	-	Ip, 2008 [58]
**Laridae**				**118**	**5568**	**2%**				
	Black-headed Gull (*Chroicocephalus ridibundus)*	2008–2011	Newfoundland and Labrador, Canada	0	1	0%	-	-	-	Huang, 2014 [39]
		2010–2011	Iceland	2	168	1%	-	-	-	Dusek, 2014 [4]
	Herring Gull (*Larus argenatus smithsonianus*)	2005	Chukchi Peninsula, Russia	0	63	0%	-	-	-	Wahlgren, 2008 [62]
		2006–2007	Alaska	0	4	0%	-	-	-	Ip, 2008 [58]
		2008–2011	Newfoundland and Labrador, Canada	13	1083	1%	H13N6 (*n* = 5), H16N3 (*n* = 3), N6 (*n* = 1)	-	-	Huang, 2014 [39]
		2010–2011	Iceland	5	121	4%	-	-	-	Dusek, 2014 [4]
	Common Gull (*Larus canus*)	2006–2007	Alaska	0	5	0%	-	-	-	Ip, 2008 [58]
		2008–2011	Newfoundland and Labrador, Canada	0	2	0%	-	-	-	Huang, 2014 [39]
		2010–2011	Iceland	0	1	0%	-	-	-	Dusek, 2014 [4]
		2015	Yukon-Kuskokwim Delta, Alaska	0	2	0%	-	-	-	Ramey, 2016 [67]
	Ring-billed Gull (*Larus delawarensis*)	2008–2011	Newfoundland and Labrador, Canada	1	21	5%	H13N6	-	-	Huang, 2014 [39]
	Lesser Black-backed Gull (*Larus fuscus*)	2010–2011	Iceland	1	96	1%	-	-	-	Dusek, 2014 [4]
	Glaucous-winged Gull (*Larus glaucescens*)	2011–2013	Izembek National Wildlife Refuge, Alaska	34	710	5%	-	-	-	Ramey, 2015 [69]
		2014	Izembek National Wildlife Refuge, Alaska	6	348	2%	-	-	-	Ramey, 2016 [68]; Reeves, 2018 [66]
	Iceland Gull (*Larus glaucoides*)	2007–2011	Greenland	1	398	0%	-	-	-	Hjulsager, 2012 [42]
		2008–2011	Newfoundland and Labrador, Canada	0	19	0%	-	-	-	Huang, 2014 [39]
		2010–2011	Iceland	1	19	5%	-	-	-	Dusek, 2014 [4]
	Glaucous Gull (*Larus hyperboreus*)	2005	Chukchi Peninsula, Russia	0	3	0%	-	-	-	Wahlgren, 2008 [62]
		2005	Pt. Barrow, Alaska	0	33	0%	-	-	-	Wahlgren, 2008 [62]
		2006–2007	Alaska	5	138	4%	-	-	-	Ip, 2008 [58]
		2007–2011	Greenland	2	110	2%	-	-	-	Hjulsager, 2012 [42]
		2008–2011	Newfoundland and Labrador, Canada	0	24	0%	-	-	-	Huang, 2014 [39]
		2010–2011	Iceland	2	98	2%	-	-	-	Dusek, 2014 [4]
		2017	Adentfjorden/Sassendalen, Norway	5	15	33%				Lee, 2020 [11]
	Glaucous Gull x Herring Gull hybrid (*Larus hyperboreus x L. argenatus*)	2007–2011	Greenland	0	47	0%	-	-	-	Hjulsager, 2012 [42]
	Great Black-backed Gull (*Larus marinus*)	2008–2011	Newfoundland and Labrador, Canada	6	200	3%	H1, H13N6, H9N9	-	-	Huang, 2014 [39]
		2010–2011	Iceland	1	12	8%	-	-	-	Dusek, 2014 [4]
		2010–2011	Iceland	4	38	11%	-	-	-	Dusek, 2014 [4]
		2021	St. John’s, Newfoundland and Labrador, Canada	1	-	-	H5N1 (HP; *n* = 1) (tissue sample)	-	-	Caliendo, 2022 [78]
	Miscellaneous Gulls (*Larus spp.*)	2007–2011	Greenland	0	747	0%	-	-	-	Hjulsager, 2012 [42]
	Aleutian Tern (*Onychoprion aleuticus*)	2006–2007	Alaska	1	302	0%	-	-	-	Ip, 2008 [58]
	Common Tern (*Sterna hirundo*)	2008–2011	Newfoundland and Labrador, Canada	0	21	0%	-	-	-	Wille, 2014 [57]
	Arctic Tern (*Sterna paradisaea*)	2006–2007	Alaska	0	1	0%	-	-	-	Ip, 2008 [58]
		2007–2011	Greenland	0	165	0%	-	-	-	Hjulsager, 2012 [42]
		2008–2011	Newfoundland and Labrador, Canada	0	9	0%	-	-	-	Wille, 2014 [57]
		2015	Yukon-Kuskokwim Delta, Alaska	1	1	100%	-	-	-	Ramey, 2016 [67]
	Black-legged Kittiwake (*Rissa tridactyla*)	2005	Chukchi Peninsula, Russia	0	1	0%	-	-	-	Wahlgren, 2008 [62]
		2006–2007	Alaska	0	2	0%	-	-	-	Ip, 2008 [58]
		2007–2011	Greenland	0	207	0%	-	-	-	Hjulsager, 2012 [42]
		2008–2009	Hornøya, Norway	25	200	13%	H4: 1/200 (0.5%)	57/98 (58.2%)	H13: 18/48 (37.5%); H16: 39/48 (81.3%)	Tønnessen, 2011 [53]
		2008–2011	Newfoundland and Labrador, Canada	0	109	0%	-	-	-	Wille, 2014 [57]
		2014–2017	Kongsfjorden, Norway	-			-	9/53 (17.0%)	-	Lee, 2020 [11]
	Sabine’s Gull (*Xema sabini*)	2006–2007	Alaska	0	1	0%	-	-	-	Ip, 2008 [58]
		2007–2011	Greenland	0	22	0%	-	-	-	Hjulsager, 2012 [42]
		2015	Yukon-Kuskokwim Delta, Alaska	1	1	100%	-	-	-	Ramey, 2016 [67]
**Motacillidae**				**0**	**308**	**0%**				
	Yellow Wagtail (*Motacilla flava*)	2005	Chukchi Peninsula, Russia	0	2	0%	-	-	-	Wahlgren, 2008 [62]
	American Pipit (*Anthus rubescens*)	2006–2007	Alaska	0	2	0%	-	-	-	Ip, 2008 [58]
	Eastern Yellow Wagtail (*Motacilla tschutschensis*)	2006–2007	Alaska	0	304	0%	-	-	-	Ip, 2008 [58]
**Paridae**				**0**	**1**	**0%**				
	Black-capped Chickadee (*Poecile atricapillus*)	2006–2007	Alaska	0	1	0%	-	-	-	Ip, 2008 [58]
**Parulidae**				**0**	**135**	**0%**				
	Bay-breasted Warbler (*Dendroica castanea*)	2006–2007	Alaska	0	1	0%	-	-	-	Ip, 2008 [58]
	Yellow-rumped Warbler (*Dendroica coronata*)	2006–2007	Alaska	0	11	0%	-	-	-	Ip, 2008 [58]
	Yellow Warbler (*Dendroica petechia*)	2006–2007	Alaska	0	7	0%	-	-	-	Ip, 2008 [58]
	Blackpoll Warbler (*Dendroica striata*)	2006–2007	Alaska	0	3	0%	-	-	-	Ip, 2008 [58]
	Northern Waterthrush (*Seiurus noveboracensis*)	2006–2007	Alaska	0	27	0%	-	-	-	Ip, 2008 [58]
	Wilson’s Warbler (*Wilsonia pusilla*)	2006–2007	Alaska	0	42	0%	-	-	-	Ip, 2008 [58]
	Orange-crowned Warbler (*Vermivora celata*)	2006–2007	Alaska	0	44	0%	-	-	-	Ip, 2008 [58]
**Phalacrocoracidae**				**0**	**30**	**0%**				
	European Shag (*Phalacrocorax aristotelis*)	2010–2011	Iceland	0	6	0%	-	-	-	Dusek, 2014 [4]
	Great Cormorant (*Phalacrocorax carbo*)	2007–2011	Greenland	0	6	0%	-	-	-	Hjulsager, 2012 [42]
		2010–2011	Iceland	0	2	0%	-	-	-	Dusek, 2014 [4]
	Pelagic Cormorant (*Phalacrocorax pelagicus*)	2005	Chukchi Peninsula, Russia	0	11	0%	-	-	-	Wahlgren, 2008 [62]
		2006–2007	Alaska	0	3	0%	-	-	-	Ip, 2008 [58]
	Red-faced Cormorant (*Phalacrocorax urile*)	2006–2007	Alaska	0	2	0%	-	-	-	Ip, 2008 [58]
**Phasianidae**				**1**	**15**	**7%**				
	Willow Ptarmigan (*Lagopus lagopus*)	2015	Yukon-Kuskokwim Delta, Alaska	0	3	0%	-	-	-	Ramey, 2016 [67]
	Rock Ptarmigan (*Lagopus muta*)	2007–2011	Greenland	0	1	0%	-	-	-	Hjulsager, 2012 [42]
	Partridge/Sharp-tailed Grouse (*Tympanuchus phasianellus*)	2013–2014	Subarctic Ontario, Canada	1	11	9%	-	-	-	Liberda, 2017 [48]
**Picidae**				**0**	**1**	**0%**				
	Northern Flicker (*Colaptes auratus*)	2006–2007	Alaska	0	1	0%	-	-	-	Ip, 2008 [58]
**Podicipedidae**				**0**	**1**	**0%**				
	Red-necked Grebe (*Podiceps grisegena*)	2006–2007	Alaska	0	1	0%	-	-	-	Ip, 2008 [58]
**Procellariidae**				**0**	**17**	**0%**				
	Northern Fulmar (*Fulmarus glacialis*)	2006–2007	Alaska	0	1	0%	-	-	-	Ip, 2008 [58]
		2008–2011	Newfoundland and Labrador, Canada	0	4	0%	-	-	-	Wille, 2014 [57]
	Manx Shearwater (*Puffinus puffinus*)	2008–2011	Newfoundland and Labrador, Canada	0	12	0%	-	-	-	Wille, 2014 [57]
**Regulidae**				**0**	**2**	**0%**				
	Ruby-crowned Kinglet (*Regulus calendula*)	2006–2007	Alaska	0	2	0%	-	-	-	Ip, 2008 [58]
**Scolopacidae**				**17**	**4333**	**0.4%**				
	Ruddy Turnstone (*Arenaria interpres*)	2005	Chukchi Peninsula, Russia	0	1	0%	-	-	-	Wahlgren, 2008 [62]
		2006–2007	Alaska	0	30	0%	-	-	-	Ip, 2008 [58]
		2010–2011	Iceland	0	64	0%	-	-	-	Dusek, 2014 [4]
		2012	Iceland	1	68	2%	-	51/60 (85%)	-	Hall, 2014 [77]
		2013	Iceland	1	88	1%	-	46/70 (66%)	-	Hall, 2014 [77]
	Black Turnstone (*Arenaria melanocephala*)	2006–2007	Alaska	0	2	0%	-	-	-	Ip, 2008 [58]
		2015	Yukon-Kuskokwim Delta, Alaska	0	3	0%	-	-	-	Ramey, 2016 [67]
	Sharp-tailed Sandpiper (*Calidris acuminate*)	2006–2007	Alaska	0	225	0%	-	-	-	Ip, 2008 [58]
		2015	Yukon-Kuskokwim Delta, Alaska	0	3	0%	-	-	-	Ramey, 2016 [67]
	Baird’s Sandpiper (*Calidris bairdii*)	2006–2007	Alaska	0	1	0%	-	-	-	Ip, 2008 [58]
	Sanderling (*Calidris alba*)	2010–2011	Iceland	0	129	0%	-	-	-	Dusek, 2014 [4]
		2012	Iceland	1	297	0%	-	1/48 (2.1%)	-	Hall, 2014 [77]
		2013	Iceland	1	388	0%	-	0/1 (0%)	-	Hall, 2014 [77]
	Dunlin (*Calidris alpina*)	2005	Chukchi Peninsula, Russia	0	22	0%	-	-	-	Wahlgren, 2008 [62]
		2005	Pt. Barrow, Alaska	1	1	100%	-	-	-	Wahlgren, 2008 [62]
		2006–2007	Alaska	2	897	0%	-	-	-	Ip, 2008 [58]
		2010–2011	Iceland	0	2	0%	-	-	-	Dusek, 2014 [4]
		2012	Iceland	0	3	0%	-	0/3 (0%)	-	Hall, 2014 [77]
		2013	Iceland	0	12	0%	-	-	-	Hall, 2014 [77]
		2015	Yukon-Kuskokwim Delta, Alaska	1	9	11%	-	-	-	Ramey, 2016 [67]
	Red Knot (*Calidris canutus*)	2006–2007	Alaska	0	79	0%	-	-	-	Ip, 2008 [58]
		2010–2011	Iceland	0	1	0%	-	-	-	Dusek, 2014 [4]
		2013	Iceland	0	21	0%	-	52/114 (45.6%)	-	Hall, 2014 [77]
	Stilt Sandpiper (*Calidris himantopus*)	2006–2007	Alaska	0	3	0%	-	-	-	Ip, 2008 [58]
	Purple Sandpiper (*Calidris maritima*)	2010–2011	Iceland	0	2	0%	-	-	-	Dusek, 2014 [4]
		2013	Iceland	0	6	0%	-	0/1 (0%)	-	Hall, 2014 [77]
	Western Sandpiper (*Calidris mauri*)	2005	Chukchi Peninsula, Russia	0	75	0%	-	-	-	Wahlgren, 2008 [62]
		2005	Pt. Barrow, Alaska	0	6	0%	-	-	-	Wahlgren, 2008 [62]
		2006–2007	Alaska	0	128	0%	-	-	-	Ip, 2008 [58]
	Pectoral Sandpiper (*Calidris melanotos*)	2005	Chukchi Peninsula, Russia	0	1	0%	-	-	-	Wahlgren, 2008 [62]
		2006–2007	Alaska	0	580	0%	-	-	-	Ip, 2008 [58]
		2015	Yukon-Kuskokwim Delta, Alaska	1	16	6%	-	-	-	Ramey, 2016 [67]
	Rock Sandpiper (*Calidris ptilocnemis*)	2005	Chukchi Peninsula, Russia	0	1	0%	-	-	-	Wahlgren, 2008 [62]
		2006–2007	Alaska	0	173	0%	-	-	-	Ip, 2008 [58]
		2015	Yukon-Kuskokwim Delta, Alaska	0	1	0%	-	-	-	Ramey, 2016 [67]
	Semipalmated Sandpiper (*Calidris pusilla*)	2005	Chukchi Peninsula, Russia	0	1	0%	-	-	-	Wahlgren, 2008 [62]
		2005	Pt. Barrow, Alaska	0	5	0%	-	-	-	Wahlgren, 2008 [62]
		2006–2007	Alaska	0	212	0%	-	-	-	Ip, 2008 [58]
	Temminck’s Stint (*Calidris temminckii*)	2005	Chukchi Peninsula, Russia	0	2	0%	-	-	-	Wahlgren, 2008 [62]
	Spoon-billed Sandpiper (*Eurynorhynchus pygmeus*)	2005	Chukchi Peninsula, Russia	0	1	0%	-	-	-	Wahlgren, 2008 [62]
	Common Snipe (*Gallinago gallinago*)	2010–2011	Iceland	0	1	0%	-	-	-	Dusek, 2014 [4]
	Wilson’s Snipe (*Gallinago delicata*)	2006–2007	Alaska	0	9	0%	-	-	-	Ip, 2008 [58]
		2015	Yukon-Kuskokwim Delta, Alaska	0	1	0%	-	-	-	Ramey, 2016 [67]
	Long-billed Dowitcher (*Limnodromus scolopaceus*)	2006–2007	Alaska	0	165	0%	-	-	-	Ip, 2008 [58]
		2015	Yukon-Kuskokwim Delta, Alaska	1	3	33%	-	-	-	Ramey, 2016 [67]
	Bar-tailed Godwit (*Limosa lapponica*)	2006–2007	Alaska	3	210	1%	-	-	-	Ip, 2008 [58]
		2015	Yukon-Kuskokwim Delta, Alaska	1	14	7%	-	-	-	Ramey, 2016 [67]
	Whimbrel (*Neumenius phaeopus*)	2006–2007	Alaska	0	1	0%	-	-	-	Ip, 2008 [58]
	Bristle-thighed Curlew (*Neumenius tahitiensis*)	2006–2007	Alaska	0	9	0%	-	-	-	Ip, 2008 [58]
	Grey Phalarope (*Phalaropus fulicaria*)	2005	Chukchi Peninsula, Russia	0	8	0%	-	-	-	Wahlgren, 2008 [62]
		2005	Pt. Barrow, Alaska	0	5	0%	-	-	-	Wahlgren, 2008 [62]
	Red Phalarope (*Phaloropus fulicarius*)	2006–2007	Alaska	0	179	0%	-	-	-	Ip, 2008 [58]
	Red-necked Phalarope (*Phalaropus lobatus*)	2005	Chukchi Peninsula, Russia	0	3	0%	-	-	-	Wahlgren, 2008 [62]
		2005	Pt. Barrow, Alaska	0	5	0%	-	-	-	Wahlgren, 2008 [62]
		2006–2007	Alaska	0	38	0%	-	-	-	Ip, 2008 [58]
		2015	Yukon-Kuskokwim Delta, Alaska	3	23	13%	-	-	-	Ramey, 2016 [67]
	Ruff (*Philomachus pugnax*)	2006–2007	Alaska	0	6	0%	-	-	-	Ip, 2008 [58]
	Greater Yellowlegs (*Tringa melanoleuca*)	2006–2007	Alaska	0	1	0%	-	-	-	Ip, 2008 [58]
	Common Redshank (*Tringa tetanus*)	2010–2011	Iceland	0	1	0%	-	-	-	Dusek, 2014 [4]
	Buff-breasted Sandpiper (*Tryngites subruficollis)*	2006–2007	Alaska	0	92	0%	-	-	-	Ip, 2008 [58]
**Slyvioidea**				**0**	**769**	**0%**				
	Arctic Warbler (*Phylloscopus borealis*)	2006–2007	Alaska	0	769	0%	-	-	-	Ip, 2008 [58]
**Strigidae**				**0**	**2**	**0%**				
	Short-eared Owl (*Asio flammeus*)	2006–2007	Alaska	0	1	0%	-	-	-	Ip, 2008 [58]
	Snowy Owl (*Nyctea scandiaca*)	2005	Pt. Barrow, Alaska	0	1	0%	-	-	-	Wahlgren, 2008 [62]
**Stercorariidae**				**0**	**6**	**0%**				
	Long-tailed Jaeger (*Stercorarius longicaudus*)	2006–2007	Alaska	0	2	0%	-	-	-	Ip, 2008 [58]
	Pomarine Jaeger (*Stercorarius pomarinus*)	2006–2007	Alaska	0	4	0%	-	-	-	Ip, 2008 [58]
**Sulidae**				**0**	**77**	**0%**				
	Northern Gannet (*Morus bassanus*)	2008–2011	Newfoundland and Labrador, Canada	0	76	0%	-	-	-	Wille, 2014 [57]
		2010–2011	Iceland	0	1	0%	-	-	-	Dusek, 2014 [4]
**Turdidae**				**0**	**333**	**0%**				
	Hermit Thrush (*Catharus guttatus*)	2006–2007	Alaska	0	10	0%	-	-	-	Ip, 2008 [58]
	Gray-checked Thrush (*Catharus minimus*)	2006–2007	Alaska	0	229	0%	-	-	-	Ip, 2008 [58]
	Swainson’s Thrush (*Catharus ustulatus*)	2006–2007	Alaska	0	20	0%	-	-	-	Ip, 2008 [58]
	Varied Thrush (*Ixoreus naevius*)	2006–2007	Alaska	0	10	0%	-	-	-	Ip, 2008 [58]
	Bluethroat (*Luscina svecica*)	2005	Chukchi Peninsula, Russia	0	2	0%	-	-	-	Wahlgren, 2008 [62]
		2006–2007	Alaska	0	12	0%	-	-	-	Ip, 2008 [58]
	Northern Wheatear (*Oenanthe Oenanthe*)	2006–2007	Alaska	0	10	0%	-	-	-	Ip, 2008 [58]
		2007–2011	Greenland	0	29	0%	-	-	-	Hjulsager, 2012 [42]
	American Robin (*Turdus migratorius*)	2006–2007	Alaska	0	11	0%	-	-	-	Ip, 2008 [58]
**Tyrannidae**				**0**	**5**	**0%**				
	Alder Flycatcher (*Empidonax alnorum*)	2006–2007	Alaska	0	5	0%	-	-	-	Ip, 2008 [58]

* Hemagglutination inhibition (HI) antibodies detected against any H5 antigen, including low pathogenic avian influenza (LPAI) H5 and highly pathogenic avian influenza (HPAI) H5. ** Varying rates of subtype-specific antibodies detected for 14 HA subtypes. See paper for data, reported by year of sampling. *** Prevalence reported, but not by species and/or location of sampling.

### 3.5. Marine and Terrestrial Mammals in the Arctic Are Exposed to IAVs, despite Major Gaps in Surveillance

IAVs can infect a range of wildlife beyond avian reservoir hosts, including marine and terrestrial mammals; however, most infections in mammals are presumed to be the result of predation of infected birds or through contact with environmental matrices contaminated by infected excreta [40,79]. Arctic foxes, for example, frequently predate nesting areas where they feed on eggs and juvenile and adult birds [80,81,82]. Consumption of infected eggs or hatched birds can cause infection in mammals [40]. Additionally, terrestrial mammals that graze around nesting areas may be exposed to excreta or contaminated plants and water sources [40,79]. Though most often infections in marine and terrestrial mammals are asymptomatic, IAVs have also caused mass die-offs, particularly among pinnipeds [45,79].

Compared to avian-focused surveillance studies, peer-reviewed data on IAVs among mammalian hosts in Arctic regions are sparse. A total of 2899 individual mammals were sampled in Arctic regions, representing a broad diversity of species (Table 2). The most sampled species were seals of the family Phocidae (1621/2899, or 55.9% of total sampled species). Peer-reviewed reports of IAVs in mammalian species were collected from several but not all geographic Arctic regions, including the Western European Arctic (Norway (*n* = 1)), Russian Arctic (*n* = 1), Canadian Arctic (*n* = 3), and Alaskan Arctic (*n* = 6). All studies on mammalian hosts of IAVs in Arctic regions included in this review measured the serological presence of antibodies to IAVs and no study sampled for detection of virus, although several studies from other global regions have detected IAVs in marine mammals [83]. In most studies, serum samples were screened for antibodies using commercially available bELISA assays targeting the IAV nucleoprotein. Few studies measured for subtype-specific antibodies through HI assays.

Pooled seroprevalence within the Arctic region indicated 9.29% and 1.69% among marine and terrestrial mammals, respectively. There were not significant differences in seroprevalence between marine and terrestrial mammals (*p* = 0.3173) or by Arctic region groupings (*p* = 0.4060). In addition to a wide range of seroprevalence rates between species, within-species differences in seroprevalence were also documented. In ringed seals, for example, researchers have documented a wide range of exposure rates, from 83.3% in the Russian Arctic to just 3.1% and 2.5% in Alaska and the Canadian Arctic, respectively [44,45,84]. Subtype-specific antibodies were measured for several host species, including ringed seals in Alaska (H3N2, H3N3, H7N7) and the central Russian Arctic (H3N2, H7N7) and Baikal seals in the central Russian Arctic (H3N2) [44,84]. In total, antibodies against three unique HA-NA subtypes and three HA subtypes were detected. Though exposure among marine mammals is generally thought to be avian in origin, serological evidence from the Russian Arctic suggests that certain infections in Baikal and ringed seals may be of human origin [44]. Several unusual mortality events linked to IAV infection have occurred in northern temperate regions, such as the Northeast USA and the Skagerrak and Kattegat Seas off western Sweden [16,79,83].

**Table 2 viruses-14-01531-t002:** Seroprevalence rates of IAVs among mammals in Arctic regions.

Family Taxa	Host Species	Sampling Year(s)	Location	Prevalence				Seroprevalence		Author, Year
				Positive Samples	Total Samples	Prevalence %	Virus Subtypes Identified	*n*/*N* (%)	Subtype-Specific Antibodies Identified, *n*/*N* (%)	
**Balaenidae**				**0**	**4**	**0%**				
	Bowhead whale (*Balaena mysticetus*)	1984–1998	Canadian Arctic	0	4	0%	-	-	-	Nielsen, 2001 [45]
**Canidae**				**1**	**231**	**0.43%**				
	Arctic fox (*Vulpes lagopus*)	2012–2016	Alaska	1	231	0.43%	-	-	-	Van Hemert, 2019 [40]
**Cervidae**				**0**	**46**	**0.00%**				
	Caribou (*Rangifer tarandus*)	2012–2016	Alaska	0	46	0%	-	-	-	Van Hemert, 2019 [40]
**Monodontidae**				**5**	**494**	**1.01%**				
	Beluga whale (*Delphinapterus leucas*)	1984–1998	Canadian Arctic	5	418	1.20%	-	-	-	Nielsen, 2001 [45]
	Narwhal (*Monodon monoceros*)	1984–1998	Canadian Arctic	0	76	0%	-	-	-	Nielsen, 2001 [45]
**Odobenidae**				**8**	**248**	**3.23%**				
	Pacific walrus (*Odobenus rosmarus divergens*)	1994–1996	Aleutian Islands, Alaska	8	38	21%	-	-	H10: 8/8 (100%)	Calle, 2002 [85]
	Atlantic walrus (*Odobenus rosmarus rosmarus*)	1984–1998	Canadian Arctic	0	210	0%	-	-	-	Nielsen, 2001 [45]
**Phocidae**				**207**	**1621**	**12.77%**				
	Hooded Seal (*Cystophora cristata*)	1991–1992	Jan Mayen Island, Norway	8	92	8.70%	-	-	-	Stuen, 1994 [86]
		1994–2005	Eastern Canadian Subarctic	4	36	11%	-	-	H3: 0 animals *	Measures, 2021 [87]
									H4: 0 animals *	
									H5: 0 animals *	
	Bearded seal (*Erignathus barbatus*)	2000–2017	Alaska	0	3	0%	-	-	-	Goertz, 2019 [88]
	Grey seal (*Halichoerus grypus*)	1994–2005	Eastern Canadian Subarctic	18	80	23%	-	-	H3: 13 animals *	Measures, 2021 [87]
									H4: 0 animals *	
									H10: 5 animals *	
	Harp seal (*Phoca groenlandica*)	1991–1992	Barents Sea and Jan Mayen Island, Norway	42	183	22.90%	-	-	-	Stuen, 1994 [86]
		1994–2005	Eastern Canadian Subarctic	86	206	41.70%	-	-	H3: 72 animals *	Measures, 2021 [87]
									H4: 4 animals *	
									H10: 11 animals *	
	Ringed seal (*Phoca hispida*)	1978–1995	Alaska	1	32	3.10%	-	-	H3N2, H3N3, H7N7: 1 animal tested	Danner, 1998 [84]
		1984–1998	Canadian Arctic	23	903	2.50%	-	-	-	Nielsen, 2001 [45]
		2000–2017	Alaska	0	11	0%	-	-	-	Goertz, 2019 [88]
		2004	Central Russian Arctic	5	6	83.30%	-	-	H3N2: 4/6 (66.6%)	Ohishi, 2004 [44]
									H7N7: 1/6 (16.6%)	
	Spotted seal (*Phoca largha*)	2000–2017	Alaska	0	8	0%	-	-	-	Goertz, 2019 [88]
	Baikal seal (*Phoca sibrica*)	2004	Central Russian Arctic	2	7	28.50%	-	-	H3N2: 1/7 (14.2%)	Ohishi, 2004 [44]
	Harbor seal (*Phoca vitulina*)	1994–2005	Eastern Canadian Subarctic	18	54	33%	-	-	H3: 0 animals *	Measures, 2021 [87]
									H4: 0 animals *	
									H5: 0 animals *	
**Ursidae**				**8**	**255**	**3.14%**				
	Brown bear (*Ursus Arctos*)	2013–2016	Alaska	8	155	5.20%	-	-	-	Ramey, 2019 [89]
	Polar Bear (*Ursus maritimus*)	2012–2016	Alaska	0	100	0%	-	-	-	Van Hemert, 2019 [40]

* Denominator not reported.

### 3.6. Arctic Environment Serves as an Important Source of IAV Infection for Wildlife

There is a sizeable environmental component to IAV infection among wildlife in Arctic habitats [40,79]. Following migration to breeding and nesting grounds in Arctic regions, infected wild birds shed virus into terrestrial and aquatic environmental matrices through excreta [90]. Viable virions have been demonstrated to persist in frozen soil and sediment, water, and ice during winter months in Arctic regions [90,91]. In warmer spring months, thawing soil and ice release infectious virions, redistributing viruses among different wildlife hosts, and infecting immune naïve adult and juvenile animals [91,92].

Peer-reviewed reports of IAVs in environmentally derived samples were collected from several Arctic regions, including the North Atlantic (Iceland (*n* = 0) and Greenland (*n* = 1)), Western European Arctic (Sweden (*n* = 0), Finland (*n* = 0), and Norway (*n* = 0)), Russian Arctic (*n* = 3), Canadian Arctic (*n* = 2), and Alaska, USA (*n* = 2) (Table 3). Pooled prevalence of samples collected via the environment in Arctic regions was 3.42% overall, and included 100%, 1.36%, 3.81%, and 5.02% among samples collected in the North Atlantic, Russian Arctic, Canadian Arctic, and Alaska, USA, respectively. Between avian, mammalian, and environmental sampling efforts in the Arctic, the greatest subtype diversity was documented through environmental sampling efforts, comprising 14 unique HA-NA subtypes, which demonstrates the utility of this method for characterizing the seasonal circulation of viruses in locations of ecological importance to host species and beyond.

In the included studies, environmental samples were collected from multiple matrices, including lake water, and excreta droppings of ducks, geese, swans, and gulls. Surveillance of duck excreta in nesting areas throughout Siberia demonstrated that IAV isolation was only successful at latitudes higher than 62°, and that no virus was isolated from duck excreta samples south of 55° [43]. The same authors collected duck excreta and lake water samples in Alaska, demonstrating that most IAVs isolated were collected in areas located above 65° north latitude during the breeding season [93]. Environmental detection of IAVs above these latitudes correlates with the long-term viability of virions in low temperature lake water (weeks to months) and ice (years) [43,90,91,93,94]. Research has also demonstrated that IAVs maintained under naturally occurring conditions in surface waters in Alaska can remain infectious for more than seven months and even exceeding one year [95,96], which suggests that IAVs perpetuate in reservoir host populations, in part, due to seasonal water-borne transmission in Arctic regions [43,51,93,97]. 

**Table 3 viruses-14-01531-t003:** Detection of IAVs in environmental samples collected in Arctic regions.

Environmental Matrix	Sampling Year(s)	Location	Prevalence				Sero-Prevalence		Author, Year
			Positive Samples	Total Samples	Prevalence %	Virus Subtypes Identified	*n*/*N* (%)	Subtype-Specific Antibodies Identified, *n*/*N* (%)	
**Lake/pond water samples**			**57**	**194**	**29.38%**				
	1992–2005	Lake Hood, Alaska	1	6	16.67%	H4N6: 1/6 (16.6%)	-	-	Ito, 1995 [93]
	1992–2005	Lake Cheney, Alaska	0	6	0%	-	-	-	Ito, 1995 [93]
	1992–2005	Potter Marsh, Alaska	0	4	0%	-	-	-	Ito, 1995 [93]
	1992–2005	Westchester Lagoon, Alaska	0	1	0%	-	-	-	Ito, 1995 [93]
	1992–2005	Lake Hanger, Alaska	0	4	0%	-	-	-	Ito, 1995 [93]
	1992–2005	Fairbanks, Alaska	0	5	0%	-	-	-	Ito, 1995 [93]
	1992–2005	Big Minto Lake, Alaska	10	44	22.73%	-	-	-	Ito, 1995 [93]
	1992–2005	Mallard Lake, Alaska	1	24	4.17%	-	-	-	Ito, 1995 [93]
	1992–2005	Heart Lake, Alaska	0	5	0%	-	-	-	Ito, 1995 [93]
	1992–2005	Canvasback Lake, Alaska	0	3	0%	-	-	-	Ito, 1995 [93]
	1992–2005	Corville Delta, Alaska	0	1	0%	-	-	-	Ito, 1995 [93]
	2001–2002	Lake Edoma, Russia	1 IAV sequence characterized			-	-	-	Zhang, 2006 [94]
	2001–2002	Lake Park, Russia	83 IAV sequences characterized			-	-	-	Zhang, 2006 [94]
	2001–2002	Lake Shchychie, Russia	0 IAV sequences characterized			-	-	-	Zhang, 2006 [94]
	2005–2006	Fairbanks, Alaska (pond 1)	17	38	44.74%	H3, H12	-	-	Lang, 2008 [98]
	2005–2006	Fairbanks, Alaska (pond 2)	20	37	54.05%	H3, H8, H11, H12	-	-	Lang, 2008 [98]
	2005–2006	Fairbanks, Alaska (pond 3)	8	16	50.00%	H3, H11, H12	-	-	Lang, 2008 [98]
**Common Murre excreta samples**			**1**	**44**	**2.27%**				
	2008–2011	Newfoundland and Labrador, Canada	1	44	2.27%	-	-	-	Wille, 2014 [57]
**Duck excreta samples, species unspecified**			**142**	**5105**	**2.78%**				
	1991–1994	Lake Hood, Alaska	2	660	0.30%	H4N6: 1/317 (0.31%); H10N7: 1/317 (0.31%)	-	-	Ito, 1995 [93]
	1991–1994	Lake Cheney, Alaska	0	460	0%	-	-	-	Ito, 1995 [93]
	1991–1994	Potter March, Alaska	0	19	0%	-	-	-	Ito, 1995 [93]
	1991–1994	Westchester Lagoon, Alaska	0	134	0%	-	-	-	Ito, 1995 [93]
	1991–1994	Lake Hanger, Alaska	0	44	0%	-	-	-	Ito, 1995 [93]
	1991–1994	Fairbanks, Alaska	0	326	0%	-	-	-	Ito, 1995 [93]
	1991–1994	Nenana, Alaska	16	108	14.81%	H3N8: 14/108 (13%)	-	-	Ito, 1995 [93]
	1991–1994	Delta Junction, Alaska	0	27	0%	-	-	-	Ito, 1995 [93]
	1991–1994	Denali National Park, Alaska	0	6	0%	-	-	-	Ito, 1995 [93]
	1991–1994	Big Minto Lake, Alaska	27	382	7.07%	-	-	-	Ito, 1995 [93]
	1991–1994	Mallard Lake, Alaska	53	180	29.44%	-	-	-	Ito, 1995 [93]
	1991–1994	Heart Lake, Alaska	6	38	15.79%	-	-	-	Ito, 1995 [93]
	1991–1994	Canvasback Lake, Alaska	0	6	0%	-	-	-	Ito, 1995 [93]
	1991–1994	Corville Delta, Alaska	0	14	0%	-	-	-	Ito, 1995 [93]
	1995–1998	Petropavlovsk-Kamchatsky, Russia	0	26	0%	-	-	-	Okazaki, 2000 [43]
	1995–1998	Lake Kanycheva, Russia	0	95	0%	-	-	-	Okazaki, 2000 [43]
	1995–1998	40 Islands, Russia	0	1185	0%	-	-	-	Okazaki, 2000 [43]
	1995–1998	Koybyaysky, Russia	32	943	3.39%	H4N6: 19/819 (2.3%); H4N9: 1/819 (0.12%); H11N1: 1/819 (0.12%); H11N6: 2/819 (0.24%); H11N9: 8/819 (0.97%); H4N6: 1/124 (0.80%)	-	-	Okazaki, 2000 [43]
	1995–1998	Kemkeme National Park, Russia	0	23	0%	-	-	-	Okazaki, 2000 [43]
	1995–1998	Kharyyalah, Russia	0	146	0%	-	-	-	Okazaki, 2000 [43]
	1995–1998	Buotama, Russia	0	51	0%	-	-	-	Okazaki, 2000 [43]
	1995–1998	Yakutsk, Russia	6	232	2.59%	H3N8: 5/62 (8.1%); H13N6: 1/170 (0.58%); H1N1: 1/170 (0.58%); H5N3: 1/170 (0.58%); H5N4 (LP): 1/170 (0.58%); H6N7: 1/170 (0.58%); H8N1: 1/170 (0.58%); H8N3: 1/170 (0.58%)	-	-	Okazaki, 2000 [43]
**Gull excreta samples**			**12**	**317**	**3.79%**				
	1991–1994	Lake Hood, Alaska	0	3	0%	-	-	-	Ito, 1995 [93]
	1991–1994	Potter Marsh, Alaska	0	4	0%	-	-	-	Ito, 1995 [93]
	1991–1994	Fairbanks, Alaska	0	3	0%	-	-	-	Ito, 1995 [93]
	1995–1998	40 Islands, Russia	0	10	0%	-	-	-	Okazaki, 2000 [43]
	2008–2011	Newfoundland and Labrador, Canada	10	295	3.39%	-	-	-	Huang, 2014 [39]
	2010	Nuuk, Greenland	1 sample *			-	-	-	Hjulsager, 2012 [42]
	2016	Kamchatka, Russia	1 sample *			-	-	-	Marchenko, 2018 [99]
**Goose excreta samples**			**4**	**794**	**0.50%**				
	1991–1994	Lake Hood, Alaska	4	238	1.68%	-	-	-	Ito, 1995 [93]
	1991–1994	Lake Cheney, Alaska	0	196	0%	-	-	-	Ito, 1995 [93]
	1991–1994	Potter Marsh, Alaska	0	130	0%	-	-	-	Ito, 1995 [93]
	1991–1994	Westchester Lagoon, Alaska	0	2	0%	-	-	-	Ito, 1995 [93]
	1991–1994	Big Minto Lake, Alaska	0	9	0%	-	-	-	Ito, 1995 [93]
	1991–1994	Mallard Lake, Alaska	0	3	0%	-	-	-	Ito, 1995 [93]
	1991–1994	Corville Delta, Alaska	0	16	0%	-	-	-	Ito, 1995 [93]
	1991–1994	Prudhoe Bay, Alaska	0	69	0%	-	-	-	Ito, 1995 [93]
	1995–1998	40 Islands, Russia	0	126	0%	-	-	-	Okazaki, 2000 [43]
	1995–1998	Kemkeme National Park, Russia	0	5	0%	-	-	-	Okazaki, 2000 [43]
**Shorebird excreta samples**			**0**	**31**	**0%**				
	1991–1994	Fairbanks, Alaska	0	7	0%	-	-	-	Ito, 1995 [93]
	1995–1998	40 Islands, Russia	0	24	0%	-	-	-	Okazaki, 2000 [43]
**Swan excreta samples**			**0**	**4**	**0%**				
	1995–1998	Kemkeme National Park, Russia	0	4	0%	-	-	-	Okazaki, 2000 [43]
	1991–1994	Corville Delta, Alaska	0	6	0%	-	-	-	Ito, 1995 [93]
**Mallard excreta samples**			**2**	**2**	**100%**				
	2007	Nuuk and Sisimiut, Greenland	2 samples *			-	-	-	Hjulsager, 2012 [42]

* Denominator not reported.

## 4. Host–Pathogen–Environmental Ecology of IAVs in Arctic Regions

### 4.1. Trans-Continental Movement of IAV via the Arctic during Migration and Breeding

While this review is specifically focused on IAVs among wildlife in Arctic regions, we recognize that many species, both avian and mammalian, migrate between Arctic and southward regions [3]. Most surveillance data included in this review are from wild aquatic birds, most of which migrate long distances along flyways which connect breeding grounds and wintering sites in southern latitudes. Biannual migrations of wild aquatic birds to Arctic regions facilitate the intermingling of multiple species from disparate global regions, driving infection dynamics, reassortment, and the global spread of infection [3,5,9,90]. It is believed that the co-evolution of LPAI viruses and wild avian hosts has resulted in infections that do not cause significant morbidity, enabling birds to transmit viruses over great distances during migration [91]. Though systematic sampling was rarely featured in the included papers, the introduction of immunologically naïve hatch-year birds at northern breeding grounds has been linked to higher seasonal prevalence during southward autumn migration than during their subsequent northward migration in the spring [13,58]. Seasonal differences in IAV prevalence are therefore governed by the annual cycle of avian hosts and the breeding ground sympatry of infected and susceptible hosts in Arctic regions [13].

Stopover and breeding locations in Arctic regions, where migratory flyways often overlap, have also been described as incursion points for the movement of IAVs between hemispheres [4,5,6,10,100]. In the North Atlantic, Iceland was recently characterized as an ecologically significant stopover and breeding site for wild migratory birds, connecting mainland Europe and Northeastern Canada for the westward movement of IAVs into North America [4,5,101]. Breeding ranges in Siberia have been described as source locations of HPAI H5N8 viruses migrating westward in 2014 to Europe, causing sizeable outbreaks in poultry and wild birds that have continued to present [6]. Southeast Asian-origin viruses isolated in north-central United States have been shown to move first through Siberia followed by the Alaskan peninsula as an entry point prior to further southward dispersal in North America [6,102]. Northern Pintails and other long-distance migrants have been implicated in the inter-continental migration of IAVs from Eurasia, transmitting to mallards with whom they share breeding habitat in Alaska, prior to ducks driving transmission to other species throughout North America [63,103]. Interactions between disparate populations of wild birds in Arctic breeding grounds, therefore, drive the dispersal of IAVs along flyways throughout the annual cycle.

### 4.2. Detection of HPAI Viruses in the Arctic Is Infrequent, but Serological Data Suggest Circulation

LPAI viruses are generally thought to be in evolutionary symbiosis with their native reservoir hosts, producing asymptomatic infections; however, HPAI viruses can cause serious clinical signs and mortality in infected hosts as well as humans [10,104,105,106]. Serological evidence has demonstrated that wild birds in Arctic regions have been exposed or are currently infected with HPAI viruses; however, HPAI viral isolates have been infrequently detected among wildlife in these regions [60,99]. In 2014, HPAI H5Nx viruses of clade 2.3.4.4 likely transmitted among wild birds in Alaska, which led to outbreaks in wild and domestic birds throughout Canada and the United States for seven months before being eradicated in poultry [10,49]. In 2016, over a year after the last local detection of HPAI virus strains, a HPAI H5N2 virus from clade 2.3.4.4 was detected in a Mallard during surveillance in Alaska, which confirmed low-frequency maintenance in wild birds in North America [73]. Despite a dearth of accessible English-language literature on IAVs in the Russian Arctic, researchers isolated a HPAI H5N8 virus in 2014 from the trachea of a duck in the Sakha Republic, a region that serves an important ecological role for birds migrating along multiple converging flyways between Southeast Asia and northern breeding grounds [70]. The same researchers described isolating and characterizing several subtypes of HPAI H5Nx viruses of clade 2.3.4.4 between 2016 and 2017 in Russia, one of which, a HPAI H5N5 virus isolated from gull excreta on Russia’s eastern Kamchatka Peninsula, was demonstrated to be virulent in laboratory mice [99]. Finally, a H9N2 IAV with mutations associated with infections in humans was isolated in Alaska in 2008 [107]. Among mammalian hosts in Arctic regions, antibodies against HPAI H7N7 virus were detected in ringed seals in Alaska and the Central Russian Arctic [44,84]. In the included studies in this review, no study reported HPAI virus infection among marine or terrestrial animals; however, HPAI viruses have been isolated from seals in Northern Germany and Denmark, red foxes in the Netherlands, and elsewhere [108,109,110]. No HPAI viruses were detected in samples collected from environmental matrices.

### 4.3. Inter-Continental Reassortment of IAVs in Arctic Regions Is Common

Due to their segmented RNA genome, IAVs can exchange gene segments through reassortment when an animal is co-infected with more than one virus, enabling the emergence of new virus strains with varying transmissibility and pathogenicity [10,39,46]. Inter-continental reassortant IAV genomes are largely driven by the long-distance pelagic phenotype of many aquatic birds, which serves to connect hemispheres and disparate host and IAV populations, as previously described [59,77]. Given that Arctic regions are stopover and breeding grounds for birds from southern regions across the globe, inter-continental reassortment of IAVs in Arctic regions is common [59]. IAVs detected in Thick-billed Murres in Greenland (2014) were comprised of fully North American lineages, suggesting an eastward flow of viruses from North America to locations in the North Atlantic, which was later demonstrated as significant via Bayesian phylodynamic modeling [5,60]. IAVs detected in common murres in Newfoundland and Labrador, Canada throughout 2008–2011 included fully Eurasian as well as North American–Eurasian reassortant lineages [39]. Similar trends have been reported in other geographies at the intersection of hemispheres, including St. Lawrence Island, Alaska, and the Alaska Peninsula [97,111]. 

During or just prior to 2014, migratory birds moved Eurasian-origin HPAI H5Nx viruses across the Bering Strait into Alaska, where reassortment with locally circulating LPAI viruses occurred before their southward dispersal in North America, which led to the most significant HPAI outbreak among wild birds and poultry in the United States and Canada until late 2021 [6,10,68]. During surveillance in Iceland between 2010 and 2011, fully North American, Eurasian, and mixed North American–Eurasian lineage IAVs were detected, the first recorded instance where entirely continental and inter-continental reassortant lineage constellations were characterized in the same geographic location [4]. Inter-species reassortment was also frequently reported among IAVs isolated from wild birds in Arctic regions. For example, the IAV gene pool in Common Murres has been described as a mosaic, comprising lineages associated with both waterfowl and gulls, and no Murre-specific lineages [59].

### 4.4. Climate Change Is Altering the Ecology of IAVs and Hosts of IAVs

Northern regions, particularly those with low seasonal temperatures and rainfall, have been predicted to have the highest probability of IAV outbreaks in wild bird populations [8]. Climate change-driven ecosystem shifts, including changes in the seasonal availability and distribution of dietary resources, have the potential to alter host–pathogen–environment dynamics in Arctic regions [46,91,112]. As seasonal temperatures in Arctic regions warm at a rate more than twice the global mean, it is unknown how changes in climactic patterns may impact the prevalence and diversity of IAV subtypes circulating in the wildlife reservoirs and hosts. Trends suggest that as climate zones in Arctic regions warm, northward shifts in population distributions, increased diversity of species in northern latitudes, and earlier spring migrations may increase the density of birds over-wintering in Arctic regions, favoring inter-species virus transmission [33,91,112]. On the other hand, given that IAVs persist in cold temperature water, soil, and ice, climate warming in Arctic breeding areas may cause decreases in abiotic virus survival, resulting in decreased transmission, infection, and global dispersal [33]. The relationship between climate change factors and wildlife–IAV dynamics in Arctic regions is complex and is deserving of increased scientific monitoring. 

## 5. Discussion

This review consolidates epidemiologic data and contributes knowledge on host, pathogen, and environmental ecology of IAVs among wildlife in the Arctic. Our search of the literature retrieved many articles; however, only 71 publications on wildlife IAV surveillance and ecology in Arctic regions met our inclusion criteria. Inclusion criteria included: articles of all study designs except experimental studies, featuring any species except humans between 1978 and 2022 in English, which include data collected from territories in Arctic regions. Though there has been a significant increase in the number of publications on this subject between 1993 and 2022, data from this region are sparse compared to other global regions [30,113]. This increase in publications may be attributed to a rise in the number of IAV outbreaks in Asia, Europe, and North America over time, enhanced surveillance systems focused on HPAI outbreak prevention, or increased funding [28,30,114,115]. Our findings demonstrate that the majority and greatest species diversity of data on wildlife IAVs in this region come from surveillance of wild avian hosts, while fewer data are available on marine and terrestrial mammals, and environmental matrices. The greatest subtype diversity, however, has been documented through environmental sampling efforts, demonstrating the utility of this approach for virus detection and subtype characterization in a particular location without the need to handle animals. Host species and temporal attribution, however, are limitations to this method. Data from the Arctic region are geographically biased, with most surveillance occurring in the Alaskan Arctic and many fewer reports from the Russian, Canadian, and Western European Arctic. 

In this review, prevalence and seroprevalence rates varied greatly across species and regions. Among avian species, the highest overall prevalence was found among dabbling ducks (*Anatidae Anatini*) (11%, *n* = 1142/10022), followed by loons (*Gaviidae*) (10%, *n* = 1/10), diving ducks (*Anatidae Aythyini*) (5%, *n* = 10/221), and auks (*Alcidae*) (4%, *n* = 150/3865) (Table 1). Gull species (*Laridae*) were found to have an overall prevalence of 2% (*n* = 118/5568) (Table 1). Prevalence estimates, however, are biased due to non-systematic sampling across seasons and species, small sample sizes, and challenges associated with surveillance of predominantly asymptomatic infections [116]. Unfortunately, most studies did not report season of sampling or hatch year (ageing categories for birds) which would serve to further contextualize both prevalence and seroprevalence data. Seroprevalence estimates are generally much higher than prevalence proportions, given the duration of antibody detection, and provide a more comprehensive overview of historic IAV exposure throughout the duration of immunological memory. For instance, among Mallards of the family *Anatidae Anatini*, we found similar seroprevalence rates in two studies: 46.4% and 46.7%, respectively [49,71]. Seroprevalence rates among Alaskan loons (*Gaviidae*) ranged from 3.4% (*n* = 1/29) among Yellow-billed Loons to 53.6% (*n* = 15/28) among Pacific Loons [40,54]. Among Black-legged Kittiwakes of the gull family *Laridae*, we found seroprevalence estimates of 58.2% (*n* = 57/98) and 17% (*n* = 9/53) from two studies conducted in the Norwegian Arctic [11,53]. These data demonstrate that wild avian species are exposed to IAVs throughout their lifetime, and prevalence estimates fail to provide the true exposure profile throughout the lifespan. Ultimately, seroprevalence studies can supplement routine virus surveillance by providing information on the cumulative incidence of exposure to IAVs throughout an animal’s lifetime; however, these methods do not determine when or where exposures occurred, but rather demonstrate that native and transient species in a given region may have introduced virus to that region in the past. 

Articles in this review also describe multiple important aspects of wildlife host, pathogen, and environmental ecology of IAVs in Arctic regions. First, the migration of wild aquatic birds provides a significant ecological mechanism for the dispersal of viruses, particularly by immuno-naïve juvenile birds migrating from northern breeding grounds in Arctic regions to southern wintering ranges [5,64,90,91,117]. Outbreaks have been demonstrated to spread along migratory flyways as well as along the perimeter of the Arctic Circle, both of which facilitate the onward inter-continental spread of LPAI and HPAI viruses to global regions with densely populated agricultural and human communities [3,5,10]. Though there is limited evidence of HPAI viruses isolated from wildlife in Arctic regions, HPAI viruses deriving from clade 2.3.4.4 have been detected among wild birds in Alaska in 2014 and Northeastern Canada in 2021, both of which caused sizeable outbreaks among wild and domestic birds throughout North America [10,78]. The frequency of HPAI incursion via Arctic zones to southern regions is currently unknown; however, given the movement of HPAI viruses in late 2021 via wild bird migration between mainland Europe and North America, most likely through North Atlantic Arctic regions [78], there is an increased need to uncover species-specific, seasonal, regional, and climate-related mechanisms of HPAI diffusion globally [2,4,5,8,40,42,57,58,118].

### 5.1. Gaps in Knowledge

Our review identified many critical gaps in knowledge about IAVs among wildlife in Arctic regions. First, no study of wildlife mammalian hosts included in this review detected virus isolates during sampling; rather, all studies evaluated IAV exposure via serological assays. Future research is necessary to understand the distribution and maintenance of IAVs in mammalian hosts, especially HPAI viruses in Arctic regions. In the absence of virus isolates collected from mammals in this region, phylogenetic reconstruction of transmission to and within mammals is limited. Second, many geographic regions within the Arctic remain unevaluated for evidence of IAVs in wildlife populations. Surveillance from vast portions of Arctic Russia and Canada have not been reported in peer-reviewed English literature; therefore, additional data on IAV exposure and infection among resident species in these areas would fill a gap in global surveillance efforts. Increased use of freely available online translation services (i.e., Google Translate) by researchers seeking to consolidate data from the non-English literature may also provide more inclusive results. Most data on IAVs in wildlife in Arctic regions come from Alaska, USA. Data from Alaska can be harnessed to generate hypotheses about IAV ecology and epidemiology in other global Arctic regions. For instance, several large wetland complexes in Alaska have been shown to serve important ecological functions for the maintenance and continental spread of IAVs [119]. Further study of wetlands in other Arctic regions will be necessary to provide a global picture of these ecologically important ecosystems. Third, much remains unknown about how changing climate regimes in Arctic regions will impact the ecology and epidemiology of IAVs and wildlife hosts. Though it is uncertain how climate warming in Arctic regions will impact wildlife host and IAV ecology, climate-aware surveillance (whereby environmental and ecological variables are collected during sampling to inform predictive models) will be an important step toward establishing baseline data with which to quantify future shifts in host–pathogen dynamics in the context of a changing Arctic climate [5,46]. Fourth, most surveillance of wild migratory birds is conducted at specific locations along migratory flyways, often breeding grounds or wintering sites where birds aggregate; however, this does not consider differences in IAV–host dynamics throughout the annual cycle [3]. Studying IAVs in wild migratory birds at multiple locations along flyways selected to provide data on spatiotemporal heterogeneity of IAV–host dynamics as well as the relative importance of Arctic and southward regions is an important next step. Fifth, the importance of interactions between biotic and abiotic mechanisms for IAV persistence and reinfection is still largely unknown [90]. Comparing environmentally-derived IAV samples to those isolated from animals from the same locations and times will enable deeper knowledge of the importance and ecology of environmental persistence and abiotic infections in animal hosts.

The following research priorities have been identified:What role do mammals serve as potential sources and sinks for inter-species transmission in Arctic regions? Are marine and/or terrestrial mammals sources of onward IAV transmission to other animals? Do Arctic mammals serve as reservoir or incidental hosts for IAVs in the region?How do wetlands throughout Arctic regions contribute to the dispersal of IAVs via waterfowl migration at regional and global scales? Are wetlands in Arctic regions associated with environmental or inter-species transmission from waterfowl to sympatric avian species and/or terrestrial mammals?What is the impact of climate change (dynamics of air temperature, sea-surface temperature, ice and permafrost melt, etc.) on IAV infection, transmission dynamics, and migratory ecology of avian and mammalian species in the Arctic?What are spatiotemporal trends (i.e., seasonal changes or geographic differences) in IAV prevalence and seroprevalence within the Arctic and how do overlaps in migratory flyways influence the global dispersal of IAVs?To what degree do environmentally derived IAV samples in the Arctic reflect the spatiotemporal dynamics of IAVs in resident and migratory birds in Arctic regions?

### 5.2. Limitations of the Study

This systematic review has several important limitations. First, our literature search included only Google Scholar and PubMed and excluded non-English language publications; therefore, our search strategy may have missed papers not featured in these databases or published in other languages, especially for studies in the Russian or Western European regions. Second, our search strategy allowed for the inclusion of articles that were not identified through our database searches but were later identified upon reviewing the bibliographies of included publications. Though we did this to increase the yield of relevant articles for this review, we may not and do not claim to have found all relevant articles that did not appear via database search. Third, though the McMaster Critical Review Form for quantitative studies [36] guided our assessment of the quality and bias of the included publications, we cannot rule out the inclusion of publication bias altogether. Fourth, our methods did not enable identification of significant differences in the prevalence of IAVs among countries with territory in the Arctic; however, this is directly linked to limited research throughout the region. Finally, given the migratory phenotype of many avian and mammalian species, we acknowledge that restricting this review to animals sampled in Arctic regions provides only a piece of the larger picture, in space and time, of an otherwise complex global ecology involving mobile and transient migratory populations. 

## 6. Conclusions

This review provides a comprehensive overview of the epidemiology and host–pathogen–environmental ecology of IAVs among wildlife in Arctic regions. While most of the literature focuses on avian species, we found considerable but limited data on IAVs derived from marine and terrestrial mammals as well as environmental matrices including bodies of water and excreta deposits in the environment. LPAI and HPAI viruses have been detected in Arctic regions and several events have spread highly consequential IAVs between Arctic and southern regions [78]. Continued and increased geographic and host-range scope for surveillance activities in the Arctic will enable predictive models of species-specific and seasonal circulation of viruses and assist in mitigating the impact of IAVs at the interface of wildlife, domestic animals, and humans.

## Figures and Tables

**Figure 1 viruses-14-01531-f001:**
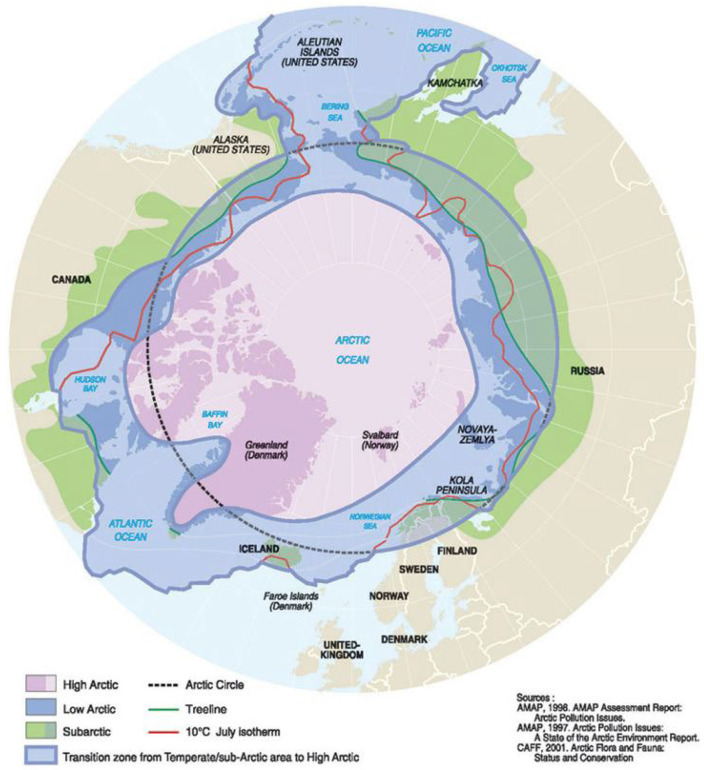
Map of Arctic region boundaries according to the Arctic Biodiversity Assessment of 2010 by the Conservation of Arctic Flora and Fauna (CAFF) working group of the Arctic Council. Reprinted with permission from Ref. [7]. 2005, Philippe Rekacewicz (UNEP/GRID-Arendal).

**Figure 2 viruses-14-01531-f002:**
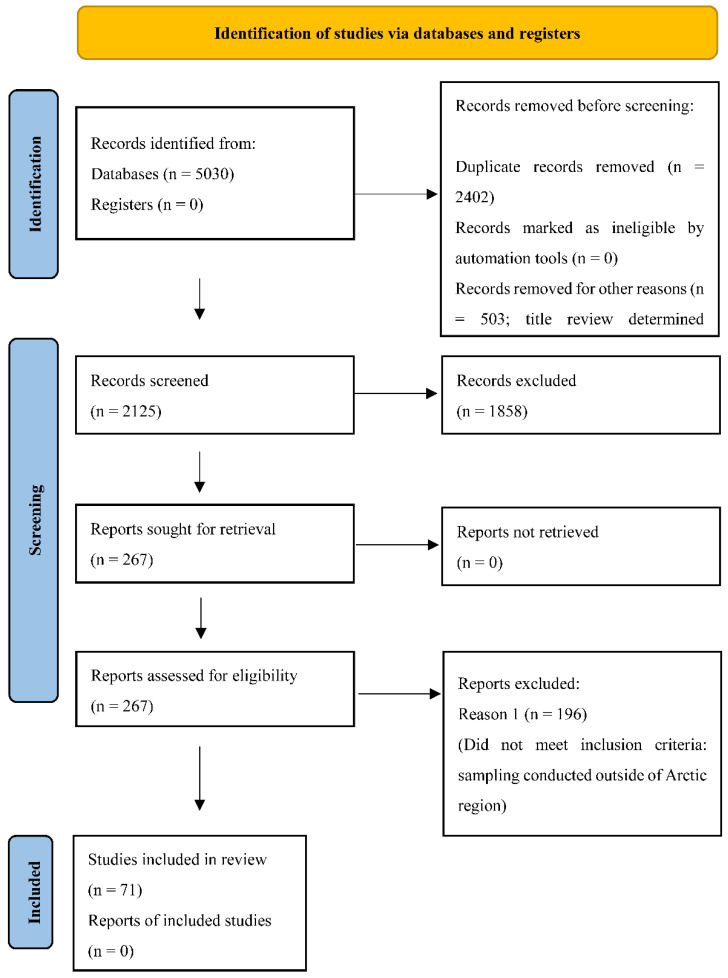
Preferred Reporting Items for Systematic Reviews and Meta-Analyses (PRISMA) flow chart of article selection.

**Figure 3 viruses-14-01531-f003:**
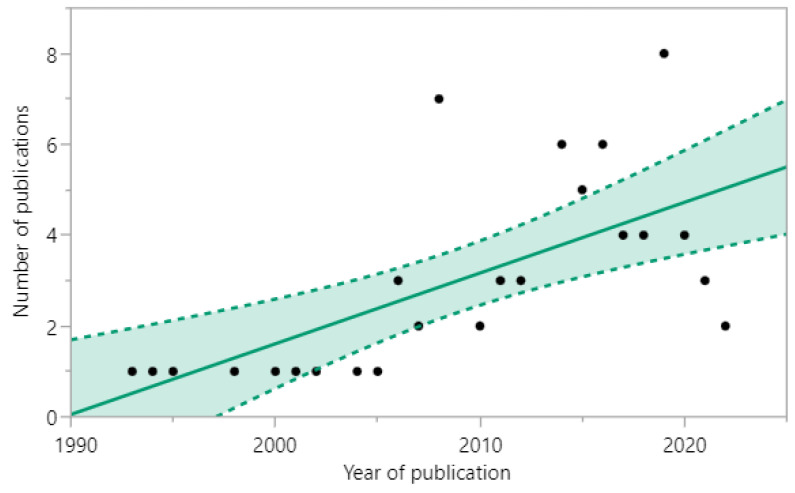
Correlation between year and number of publications. Solid line is the best fit line, dashed lines are the 95% confidence interval around the linear fit model.

**Figure 4 viruses-14-01531-f004:**
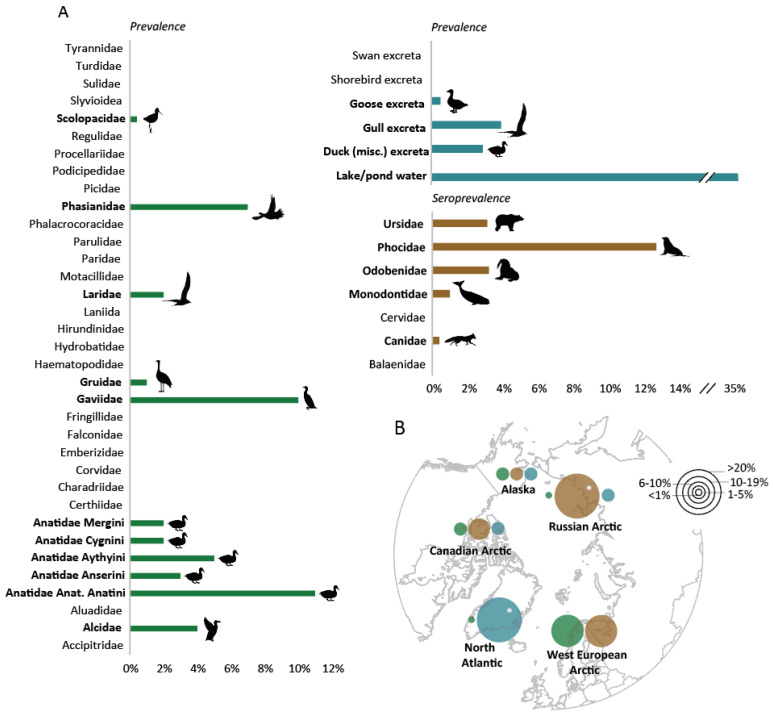
(**A**) Pooled prevalence of IAVs among wild birds (green) and environmental matrices (blue), and seroprevalence among marine and terrestrial mammals (brown), by taxonomic Family, 1978–2022. (**B**) Map of Arctic regions (clockwise from top: Russian Arctic, Western European Arctic, North Atlantic Arctic, Canadian Arctic, and Alaska, USA), where circles represent prevalence among wild birds (green), environmental matrices (blue), and seroprevalence by marine and terrestrial mammals (brown). The presence of * inside circles denotes very low sample sizes (see Table 1 for more detail). Animal icons made by Freepik from www.flaticon.com, accessed on 16 May 2022.

## Data Availability

Not applicable.

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
