# Peer review of "Epidemiology and Ecology of Influenza A Viruses among Wildlife in the Arctic"

_viruses, 2022, doi:10.3390/v14071531_

Round 1

Reviewer 1 Report

Please find a review for your consideration, attached.

Author Response

Dear Ayato Takada, Iris Zhang, and reviewers,

Thank you for your substantive and positive review of our manuscript. We look forward to publishing this manuscript in the journal Viruses. Please see below our responses to each of the reviewers’ comments in bold. We have used Track Changes to ensure the editorial team can see where edits have been made according to reviewer comments. Please also note that when responding to these reviews, we noticed a few typos and may have clarified language throughout, which are also reflected in track Changes. We appreciate the opportunity to publish this systematic review in Viruses and are very appreciative to the time and effort put forth by the reviewers and editors.

Best,

Jon Gass

__________________________________________________

Manuscript# viruses-1791408. Gass et al. Epidemiology and ecology of Influenza A viruses Among Wildlife in the Arctic: A Systematic Review

Comments to the Authors

Viruses

Manuscript# viruses-1791408

Authors: Gass et al.

Title: Epidemiology and ecology of Influenza A viruses Among Wildlife in the Arctic: A Systematic Review

Reviewer summary: This manuscript provides a systemic review of IAV ecology and epidemiology in Arctic regions, in both avian and mammalian wildlife species. This manuscript is well written and designed. This reviewer found minor issues/questions, which are highlighted below. The following comments are provided for the consideration of the authors and in support of their preparation of this manuscript. I recommend this manuscript for publication with incorporation of revisions outlined below. Thank you for the opportunity to review this body of work.

 ABSTRACT: The abstract is clear and well-written.

INTRODUCTION: The introduction is well-written. The objectives and background information here are clear, as are current knowledge gaps that authors set out to tackle.

MINOR COMMENTS: LINE 66 – remove “hemagglutinin”; already abbreviated in line 62

Thank you for spotting this oversight. We have revised the manuscript draft accordingly.

METHODS: MINOR COMMENTS

LINE 93 – remove “and subarctic regions”; “Arctic regions” was defined to include subarctic in line 43

Thank you. This has been revised accordingly.

LINE 112 – though it is implied based on “natural infections in wildlife”, I assume Galliformes were excluded from inclusion, or just domesticated species? Should this be indicated?

While we feel the original sentence is correct, we have revised it to read: “To reflect natural infections in wildlife and environmental transmission of IAVs in Arctic regions, all study designs except experimental laboratory-based studies, and all wildlife species, met criteria for inclusion.”

LINE 149 – please define what “prevalence” is based on… VI vs PCR vs direct sequencing? What is “seroprevalence” based on? bELISA (what S/N threshold?), AGID, microneutralization, hemagglutinin inhibition, etc?? If microneut or HI, are these totals reflective of a subset of previously screened bELISA(+) samples, etc?

We appreciate your interest in including this information. Prevalence is based on PCR results and seroprevalence is based on bELISA results. s/N thresholds likely varied somewhat across studies and we feel that level of detail is unnecessary to include in this review paper. The sentence has been revised to read “Prevalence (based on polymerase chain reaction (PCR) results) and seroprevalence (based on blocking enzyme-linked immunosorbent assay (bELISA) results) of IAVs were extracted from included papers and reported by species and year and combined to summarize prevalence and seroprevalence by taxonomic family.”

RESULTS / FIGURES / TABLES MAJOR COMMENTS All figures are very nicely done!

Thank you! We tried our best to make certain figures were user-friendly and interpretable.

TABLE 1 – are detected H5 subtypes LP or HP? Herring gull, GBBG, 2008-11, NL, CA – how many of each subtype? GBBG, 2021 NL, CA – 1(+) of how many collected?

Thank you for the detailed read of the table. We have made a note in the table (LP or HP) next to each reported H5 subtype for all H5 subtypes reported. The column headers for prevalence and seroprevalence now include definitions for LP: low pathogenic and HP: high pathogenic so that the reader can reference these. As for the GBBG, 2021, NL, CA, only one tissue sample was collected and analyzed, therefore we have revised the denominator and prevalence column to include “-“ as these are not applicable.

With so few subtypes actually reported for any of the detected viruses, how meaningful are any of these presumptive (+)? Assume these are just PCR hits, PCR contamination, very high threshold, and/or non-infectious viral RNA. How does this contribute to transmission/maintenance? Thresholds and testing strategies need to be defined/teased out here because is apples-to-oranges across of all these studies without that information in hand so that reader can make an informed decision about the results.

While we agree that few subtypes are reported throughout the literature included in this review, we do not draw the conclusion that these are less meaningful or inaccurate as a result. Every study approaches subtypification differently and Table 1 already contains a huge amount of information – there is little room to add any additional columns. We do agree, however, that the limited number of subtypes listed does not contribute very much to increased knowledge about transmission or maintenance. If you would prefer we remove this column altogether we can do this, however we strongly feel that this column’s inclusion is still informative and can compel the reader to visit the referenced publication to learn more about the thresholds and testing strategies used in each given publication.

There is no temporal resolution here – given that timing/seasonality is so important to IAV epidemiology, prevalence, and seroprevalence estimates reported without context of seasonality are not very informative. I see this is a major flaw in the reporting and interpretation of this compilation of data. While this is brought out a bit in discussion, I think some added detail in the table could better inform the reader.

Thank you. We agree that seasonality of IAV circulation is very important. Across all of the papers listed in Tables 1 and 3 that report prevalence data (Table 2 only reports seroprevalence data), most studies did not implement systematic sampling across seasons/years and most report prevalence data from just a single period of time. We feel that season, as reported in the included papers, is more suggestive of the time of sampling versus the seasonality of virus circulation. While we do completely understand your point, we feel that it could confuse the reader to attempt to distill season across all included studies, especially since date of sampling is so variably reported across studies. We already included the following to the Results section, page 59, lines 12-17: “Though systematic sampling across the annual cycle was rarely a feature in included papers, the introduction of immunologically naïve hatch-year birds at northern breeding grounds has been linked to higher seasonal prevalence during southward autumn migration than their subsequent northward migration in the spring [13, 58]. Seasonal differences in IAV prevalence are therefore governed by the annual cycle of avian hosts and the breeding ground sympatry of infected and susceptible hosts in Arctic regions [13].” (Please note the italicized words are newly added to make the point clearer). We then discussed the fact that most papers do not systematically sample across seasons in the Discussion on page 62, lines 133-137: “Prevalence estimates, however, are biased due to non-systematic sampling across seasons and species, small sample sizes, and challenges associated with surveillance of predominantly asymptomatic infections [119]. Unfortunately, most studies did not report season of sampling or hatch year (ageing categories for birds) which would serve to further contextualize both prevalence and seroprevalence data.” We feel that these statements sufficiently cover the role of seasonality for purposes of this systematic review, along with other statements on seasonality included in the Results and Discussion sections.

Not necessary, but possible to use the avian icons from Fig 4 in Table 1 to highlight taxa from which IAV have been detected?

This is a great idea, which we appreciate, however since not all taxa reported in the table 1 have prevalence rates above 0, there would be taxa with no bird icon and we feel this would cause confusion for the reader. We feel that Figure 4 can be reviewed and the reader can then refer to the relevant taxa in table 1 by name versus by icon.

MINOR COMMENTS LINE 240 (and elsewhere)– does journal require inclusion of Genus species at first usage?

This is a question we hope the Editor can provide an answer to.

LINE 248 – what is “viral swabbing” as a detection method? This is a collection method in my opinion, PCR or VI etc as the actual detection method?

 Yes, we agree with this and have changed the wording accordingly.

FIGURE 4 LEGEND – Should “1928” be “1978”?

Keen eye! We have changed accordingly – our mistake.

LINE 59, pg 60 – Change “among” to “and”

This change has been made.

LINE 79, pg 60 – change “just prior” (used in beginning of same sentence) to “before” or the like.

This change has been made.

LINE 81, pg 60 – change “until this year” to “2022”.

This change has been made, to read “until late 2021” as the most recent and current outbreaks started in 2021 in North America.

LINE 82, pg 60 – change “North America-Eurasian” to “North American-Eurasian”

This change has been made.

DISCUSSION LINE 117 – and any other reasons, increase in funding, e.g.?

We have revised the sentence to read: “This increase in publications may be attributed to a rise in the number of IAV outbreaks in Asia, Europe, and North America over time, enhanced surveillance systems focused on HPAI outbreak prevention, or increased funding [114, 116-118].”

LINE 138 – You are not reporting only neutralizing Ab data in table though, are you?

Correct, we have remove the word ‘neutralizing’ to better reflect this is about antibody detection and not neutralizing antibody assay results.

LINE 217 – Gap #4 – where is this coming from? Seems out of the blue related to body of manuscript?

This is an important observation. We have other papers which address this identified gap and therefore we included, but we agree to remove this gap because the review does not address this topic directly, despite pollutant exposure being related to climate change and increased risk of exposure in Arctic regions.

____________________________________________

Second review:

This manuscript is comprehensive review on IAVs of the wildlife in Arctic regions. Its overall structure and contents are well organized, and it will be helpful for readers in the related field. I have enjoyed reading the manuscript, and some minor points to improve the manuscript are as below:

- line 253, Using 1000 separator (,) is recommended.

Thank you, we have revised accordingly.

- Table 1, A typo was found. In case of Sulidae, two papers were included for prevalence analysis. One paper with 76 samples and the other with 1 sample. But, the sum is 410 in the current table. In addition, as the table is long, it is recommended to put first row with table headings together with each following parts of table.

Thank you for spotting this typo, we apologize for the oversight. We have rectified this error and the newly submitted manuscript includes revised #s resolving this error. As for the column headers, we agree completely with the sentiment, however it is out of our control how these appear in the online version of the journal. We seek advice from the journal how to deal with this large table and its user-friendliness in the online version of the journal as well as the pdf version of the manuscript. The journal may want to include this as a scroll-able table in the online version with the header frozen at the top (header, therefore, would not scroll with the contents of the table as the reader scrolls through it). We are not sure how the journal handles large tables. Please advise.

- Table 2, As the table is long, please put first row with table headings together with each following parts of table.

Please see response above.

- Line 1, page 50, …source of infection…. -> …source of IAV infection…..

Thank you, we have revised accordingly.

- Table 3, As the table is long, please put first row with table headings together with each following parts of table.

See comment above please.

- Line 10, page 59, co-evolution of LPAI -> co-evolution of LPAIVs (or LPAI viruses)

Keen eye!  Your edits are appreciated. We have revised accordingly.

- line 34, page 59, HPAI ->HPAIVs (or HPAI viruses)

This edit has been made.

- line 44, page 59, HPAI strains -> HPAI virus strains

Correction made, thank you.

- line 55, page 60, HPAI H7N7 -> HPAI H7N7 virus

Correction made, thank you.

- line 57, 58 of page 60, HPAI -> HPAIV (or HPAI virus)

Thank you – the revised sentence reads “In the included studies in this review, no study reported HPAI virus infection among marine or terrestrial animals, however HPAI viruses have been isolated from seals in Northern Germany and Denmark, and red foxes in the Netherlands, and elsewhere [109-111].”

- line 170-176 of page 62, I agree with further study about direct evidence for IAVs infection in Arctic mammals. In addition to this, I would like to suggest the authors to add a sentence about that the further study will be helpful to get knowledge whether the Arctic mammals are tending to be reservoir hosts or incidental hosts for IAVs in Arctic region. 

Thank you, we have added the sentence “Do Arctic mammals serve as reservoir or incidental hosts for IAVs in the region?”

Reviewer 2 Report

This manuscript is comprehensive review on IAVs of the wildlife in Arctic regions. Its overall structure and contents are well organized, and it will be helpful for readers in the related field. I have enjoyed reading the manuscript, and some minor points to improve the manuscript are as below:

- line 253, Using 1000 separator (,) is recommended.

- Table 1, A typo was found. In case of Sulidae, two papers were included for prevalence analysis. One paper with 76 samples and the other with 1 sample. But, the sum is 410 in the current table. In addition, as the table is long, it is recommended to put first row with table headings together with each following parts of table.

- Table 2, As the table is long, please put first row with table headings together with each following parts of table.

- Line 1, page 50, …source of infection…. -> …source of IAV infection…..

- Table 3, As the table is long, please put first row with table headings together with each following parts of table.

- Line 10, page 59, co-evolution of LPAI -> co-evolution of LPAIVs (or LPAI viruses)

- line 34, page 59, HPAI ->HPAIVs (or HPAI viruses)

- line 44, page 59, HPAI strains -> HPAI virus strains

- line 55, page 60, HPAI H7N7 -> HPAI H7N7 virus

- line 57, 58 of page 60, HPAI -> HPAIV (or HPAI virus)

- line 170-176 of page 62, I agree with further study about direct evidence for IAVs infection in Arctic mammals. In addition to this, I would like to suggest the authors to add a sentence about that the further study will be helpful to get knowledge whether the Arctic mammals are tending to be reservoir hosts or incidental hosts for IAVs in Arctic region. 

Author Response

(The authors gave the same response as above.)
